# Class-aware Domain Knowledge Fusion and Fission for Continual Test-Time Adaptation

**Jiahuan Zhou[1], Chao Zhu[1], Zhenyu Cui[1], Zichen Liu[1], Xu Zou[2]\*, Gang Hua[3]**

[1]Wangxuan Institute of Computer Technology, Peking University, Beijing 100871, China
[2]the Huazhong University of Science and Technology, Wuhan 430074,China
[3]Amazon.com, Inc, Bellevue, WA 98004, USA
jiahuanzhou@pku.edu.cn, zhuc2022@mail.sustech.edu.cn
{cuizhenyu,lzc20180720}@stu.pku.edu.cn, zx@zoux.me, ganghua@gmail.com

## Abstract

Continual Test-Time Adaptation (CTTA) aims to quickly fine-tune the model during the test phase so that it can adapt to multiple unknown downstream domain distributions without pre-acquiring downstream domain data. To this end, existing advanced CTTA methods mainly reduce the catastrophic forgetting of historical knowledge caused by irregular switching of downstream domain data by restoring the initial model or reusing historical models. However, these methods are usually accompanied by serious insufficient learning of new knowledge and interference from potentially harmful historical knowledge, resulting in severe performance degradation. To this end, we propose a class-aware domain Knowledge Fusion and Fission method for continual test-time adaptation, called KFF, which adaptively expands and merges class-aware domain knowledge in old and new domains according to the test-time data from different domains, where discriminative historical knowledge can be dynamically accumulated. Specifically, considering the huge domain gap within streaming data, a domain Knowledge FIssion (KFI) module is designed to adaptively separate new domain knowledge from a paired class-aware domain prompt pool, alleviating the impact of negative knowledge brought by old domains that are distinct from the current domain. Besides, to avoid the cumulative computation and storage overheads from continuously fissioning new knowledge, a domain Knowledge FUsion (KFU) module is further designed to merge the fissioned new knowledge into the existing knowledge pool with minimal cost, where a greedy knowledge dynamic merging strategy is designed to improve the compatibility of new and old knowledge while keeping the computational efficiency. Extensive experiments on the ImageNet-C dataset verify the effectiveness of our proposed method against other methods. The source code is available at https://github.com/zhoujiahuan1991/NeurIPS2025-KFF.

## 1 Introduction

Recently, deep neural networks have demonstrated powerful adaptation capabilities on various vision tasks [10, 14, 62, 50], but still suffer from the well-known distributional shift problem [43, 15, 22, 51, 49] between training and test data. To address this problem, Test-Time Adaptation (TTA) is proposed to adapt the test data in the target domain by using only unlabelled streaming test data [46, 57, 29, 17, 1]. Existing TTA methods have shown promising capacity to improve the generalizability of pre-trained models through self-supervised training methods [46, 39, 57, 58, 5, 55]. Despite some progress, most TTA methods merely focus on the generalizability within a single

---

\*Corresponding author

39th Conference on Neural Information Processing Systems (NeurIPS 2025).

testing domain, ignoring the multiple test scenarios that may appear from time to time in real scenarios [47, 8].

To tackle the above issue, Continual Test-Time Adaptation (CTTA) aims to exploit the unlabelled test data streams with continually changing testing domains for test-time adaptation [47, 38], as shown in Figure 1(a). The core challenge of CTTA is to adapt to the changing test data distribution by reducing error accumulation and preventing catastrophic forgetting to improve the robustness of long-term adaptation. To this end, some CTTA methods mainly preserve historical knowledge through regularization [47, 38, 45, 32] or restoration [37, 38, 54], which aim to slow down the learning of new data and correct domain style bias, thereby suppressing the impact of distributional shift problem caused by domain gaps, respectively. Unfortunately, these methods ignore the domain conflict between streaming data collected from distinct domains, which fails to fully accumulate differential domain knowledge in various test-time data. Specifically, advanced CTTA methods [33, 61] typically select and fuse parameters of historical models. Therefore, as shown in Figure 1(b), the knowledge offset between two distinct domains will inevitably disrupt the gra-

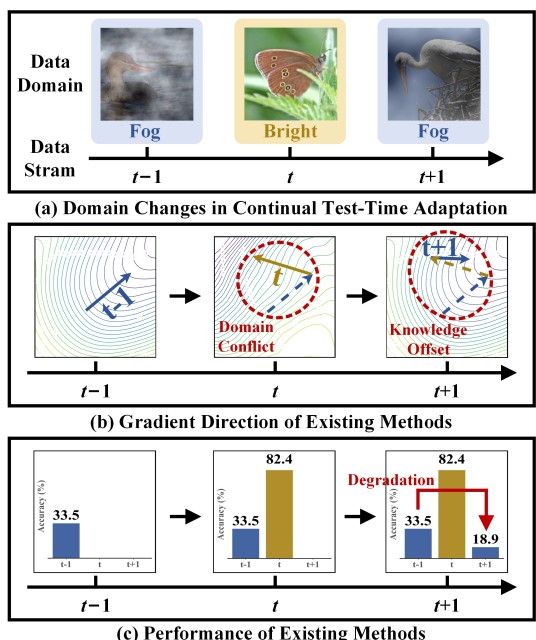

(a) Domain Changes in Continual Test-Time Adaptation

(b) Gradient Direction of Existing Methods

(c) Performance of Existing Methods

Figure 1: Existing methods [32] failed to accumulate knowledge in distinct domains, where conflicting knowledge disrupts the gradient direction during optimizing and remains a severe domain conflict problem.

dient direction during TTA optimization. Consequently, as Figure 1(c) illustrates, it not only brings an unlimited and continuously increasing storage overhead of historical models, but also compromises the discriminability due to the inevitable mixing of conflict domain knowledge. Therefore, CTTA remains a challenging issue to solve when considering the trade-off between adaptation efficiency and effectiveness.

To adapt pre-trained models to downstream tasks efficiently, prompt learning [4, 41, 20, 12] proposes to adjust a small set of parameters while keeping the pre-trained parameters fixed. However, existing prompt learning methods typically face two key challenges. For one, existing prompt learning methods [61, 11, 53] fail to separate category information and domain information, resulting in a mixture of discriminative information from different domains in the resulting prompt, which inhibits the discrimination of the adapted prompts. Second, existing prompt learning methods [6, 61] are difficult to dynamically adjust the prompting capacity according to different testing domains. Therefore, when the test domain switches irregularly, the adapted prompts typically weaken the robustness of discriminability in various domains.

Inspired by the above observations, we proposed a class-aware domain Knowledge Fusion and Fission (KFF) framework for continual test-time adaptation. To accumulate various knowledge in different domains, we designed a Knowledge FIssion (KFI) and a Knowledge FUsion (KFU) module to achieve the continual evolution of historical knowledge. Specifically, a KFI module is proposed to dynamically fission class-aware domain knowledge adapted to the current domain by evaluating the knowledge discrepancy between the current domain and the historical domains. Sequentially, a KFU module is introduced to merge the fissioned knowledge into the existing knowledge pool at a minimal cost. Among them, a greedy-based knowledge fusion strategy is proposed to achieve the fusion of various knowledge with minimal risk of old knowledge loss.

We evaluate our KFF with three common CTTA benchmarks (ImageNet-to-ImageNet-C [15], CIFAR100-to-CIFAR100C and CIFAR10-to-CIFAR10C [23]) under continual changing domains [47, 16, 6]. We further compare our KFF with state-of-the-art algorithms, including the latest advancements in CTTA and TTA fields. The experimental results show its effectiveness under various changes of testing domains. In particular, our KFF achieved 34.8% error under the ImageNet-to-ImageNet-C distributional shift case, which surpasses the previous SOTA method DPCore [61] by 5.1%.

## 2 Related Work

**Test-time Adaptation.** Test-time adaptation (TTA) aims to adapt pre-trained models to handle distribution shifts during inference, without access to source data or additional supervision [7, 12, 18, 19, 36, 29, 44]. Some methods employ self-supervised losses, such as entropy minimization [46, 39, 57, 58] or consistency maximization [5, 55], to adjust the model. Some methods involve preliminary steps to use source data: by extracting source characteristics such as statistics or features [59, 35, 40, 60], or by warming up injected parameters on source data before adaptation [12, 24, 45]. However, existing TTA methods typically assume a static target domain and fail to account for domain shifts that evolve over time [3, 13, 47]. This limitation results in challenges such as error accumulation and catastrophic forgetting, which significantly degrade model performance and adaptability during inference.

**Continual Test-time Adaptation.** Compared to TTA, continual test-time adaptation (CTTA) considers a more practical scenario in which the target domain continuously evolves. This setting exacerbates challenges like error accumulation [5] and catastrophic forgetting [47]. To address these issues, some methods such as EATA [38] and EcoTTA [32] introduce regularization strategies to mitigate error accumulation, while others like ERSK [37], RDumb [42] and CoTTA [47] utilize weight reset mechanisms to counteract catastrophic forgetting. Beyond updating the model itself, some approaches leverage a small number of parameters to incrementally learn target-domain-specific knowledge (*e.g.,* VDP [11], SVDP [53], and ViDA [27]). However, these methods struggle to retain domain-specific knowledge over time, resulting in poor performance when previously encountered domains reappear. More recently, DPCore [61] attempts to preserve historical domain knowledge and dynamically compose it during inference. While it effectively retains past domain information, it applies all previously stored knowledge to each new test batch without considering potential domain conflicts, which may lead to suboptimal performance.

**Prompt Learning.** Prompt learning is initially introduced in natural language processing (NLP) [4, 41] as a means of using learnable prompt tokens to better adapt pre-trained models to downstream tasks. Inspired by its success, researchers have extended this approach to computer vision [20, 12, 48, 52, 26, 31], achieving competitive results. Motivated by this, several methods have explored the integration of prompt learning into TTA [12, 56, 30] and CTTA scenarios [53, 11, 40, 21]. For instance, VDP [11] and SVDP [53] propose self-training models that adapt learnable visual prompts to dynamically changing domains. Other methods, such as DePT [12], CPT4 [21] and DPCore [61] introduce learnable prompts into Vision Transformers (ViTs), enhancing their ability to handle complex visual inputs and improving performance in TTA and CTTA settings. However, existing prompt-based TTA/CTTA approaches focus primarily on prompting for knowledge at the domain level, often overlooking the shared class-level information across domains, which could further enhance generalization.

## 3 Method

### 3.1 Problem Formulation and Notations

**CTTA Problem Formulation.** We focus on continual test-time adaptation here, where the target distribution differs from the source distribution and is not static. The training data are from the source domain $\mathcal{D}_S = \{Y_S, X_S\}$, and the test data are from different domain distributions dominated as $\mathcal{D}_T = \{X_T\}_{T=1}^N$, where $N$ represents the number of potential target domains, which is unknown and can be infinite or repeating. The model $f_\theta$ encounters test batches $\mathcal{B}_j^T = \{x_t\}_{t=0}^b, x_t \in X_T$ of batch size $b$ in an online manner, which means it will only meet one batch $\mathcal{B}_t$ at test time $t$. The entire process cannot access any source domain data and can only access the target domain data once. With continually changing domains, our goal is to adapt the pre-trained model to target domains and maintain the ability of the model on historical domain distributions.

**Vision Transformers(ViTs).** We focus on ViTs for their outstanding representation learning powers. A ViTs $f$ can be decomposed into a feature extractor $\phi : \mathcal{X} \to \mathcal{Z}$ and a classifier head $h : \mathcal{Z} \to \mathcal{Y}$,

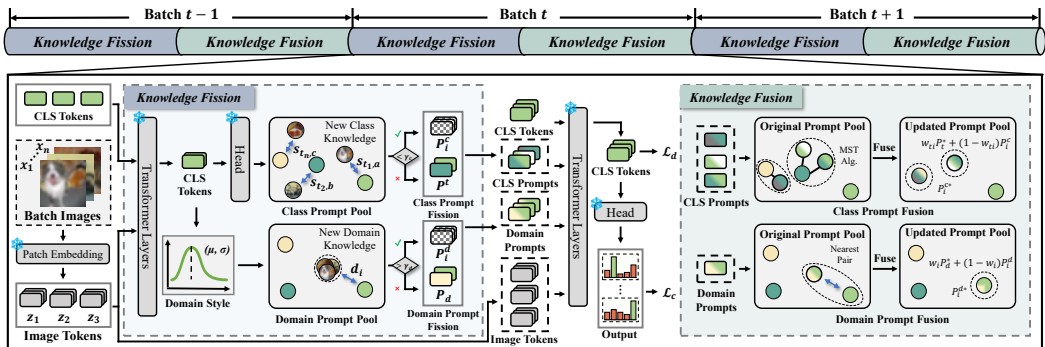

Figure 2: The Overall pipeline of our KFF with Knowledge Fission(KFI) and Knowledge Fusion(KFU). For each test batch, the KFI module dynamically fissions a domain prompt and several class prompts to adjust the source model. Sequentially, KFU merges the fissioned prompts after optimizing, achieving the balance between effectiveness and efficiency.

such that $f = \phi \circ h$. Let $z_0 \in \mathcal{Z}$ represents the classification token in $\mathcal{Z}$, the standard prediction process follows:

$$\begin{aligned} \mathcal{Z} &= \phi(\mathcal{X}), \; y = h(z_0) \\ \hat{y} &= \text{softmax}(y) \end{aligned}. \tag{1}$$

## 3.2 Overview of the Proposed KFF

As shown in Figure 2, our proposed method mainly contains two key modules: Knowledge Fission and Knowledge Fusion. Given input test batch $\mathcal{B}_j^T$ from domain $\mathcal{D}_T$, we choose or fission class prompts $\{\mathcal{P}^t\}_{t=0}^b$ for every single test sample in $\{x_t\}_{t=0}^b$ and domain prompts $\mathcal{P}_j^T$ for the whole test batch $\mathcal{B}_j^T$ and get prompts $\mathcal{P} = \{\mathcal{P}_j^T, \{\mathcal{P}^t\}_{t=0}^b\}$, then we apply it to the pre-trained ViT model $\theta_s$ to get the predicted category $\hat{y} = f(\mathcal{B}_j^T; \theta_s, \mathcal{P})$ of input test batch $\mathcal{B}_j^T$. To optimize model performance, prompts are trained using a bi-level loss function that integrates both batch-level domain alignment ($\mathcal{L}_d$) and instance-level classification entropy ($\mathcal{L}_c$). We adopt domain alignment for batch-level loss computation, leveraging its established effectiveness and computational efficiency [2, 34]. Specifically, the domain alignment loss $\mathcal{L}_d$ measures the discrepancy between the source domain and the current domain by calculating the combined Euclidean distance of their feature means and standard deviations:

$$\mathcal{L}_d = \left\| \mu^s - \mu_j^T(\mathcal{P}) \right\|_2 + \alpha \left\| \sigma^s - \sigma_j^T(\mathcal{P}) \right\|_2. \tag{2}$$

Notably, while this domain alignment distance calculation requires source data, labels are not needed, as we only perform marginal distribution alignment. And approximately 300 unlabelled source examples are enough for stable performance [61]. At the instance level, we minimize the prediction entropy using:

$$\mathcal{L}_c = \frac{1}{b} \sum_{t=0}^b \mathcal{H}(\hat{y}_t). \tag{3}$$

The model then learned $\mathcal{P}^*$ from $\mathcal{P}$ by minimizing $\mathcal{L} = \mathcal{L}_d + a\mathcal{L}_c$ and use the learned prompts to update the two prompt pools, enabling adaptive knowledge accumulation across batches. After the update, a fusion step is carried out to control the size of the prompt pool, i.e. $N_c$ and $N_d$, and retain historical knowledge, balancing computational efficiency with the preservation of key information for improved generalization and performance.

## 3.3 Knowledge Fission Module

Noticed that there might be performance degradation due to domain conflicts, as shown in Figure 1, we proposed a knowledge fission strategy to prevent the current test batch from being influenced by the conflict historical knowledge as follows:

**Class Knowledge Fission.**    In this submodule, we aim to handle the knowledge fission at class level. To achieve this, we use the cosine similarity $s_{t,i} = \text{sim}(\tilde{y}_t, y_i)$ between pseudo labels $\tilde{y}_t$ and prompt keys $y_i$ to evaluate prompt $\mathcal{P}_i$ for every test sample in the whole batch. Specifically, we first extract pseudo-labels $\tilde{y}_t$ without relying on any prompt, which allows us to obtain an initial understanding of the samples' characteristics without the influence of existing prompts. Then, we evaluate $\mathcal{P}_i$ by $s_{t,i}$, which helps us to determine how well a prompt aligns with the pseudo-label of a test sample. The prompts that $s_{t,i} > \gamma_c$ will be selected as candidates and will be used in a weighted manner:

$$\mathcal{P}^t = \sum_{i=0}^{N_c} w_i \mathcal{P}_i^c, \ w_i = \frac{\exp(s_{t,i}/\tau_c)}{\sum_{i=0}^{N_c} \exp(s_{t,i}/\tau_c)}. \tag{4}$$

If no candidates were selected for $x_t$, which means the model finds a new class which has not been seen before and is not similar to any seen classes, the model will fission a new prompt $\mathcal{P}^t$ for the test sample. This new prompt is designed to capture the unique characteristics of the new class. It is worth mentioning that at the initial state when the pool is empty, we will fission a prompt for every single test sample. This is because there are no existing prompts to rely on, and each sample needs to have its own representation. The class prompt $\mathcal{P}_c$ is concatenated by $\mathcal{P}^t$ of each test sample $x_t$ and will be used for prediction and learning:

$$\mathcal{P}_c = \left[ \mathcal{P}^0, \mathcal{P}^1, \ldots, \mathcal{P}^b \right]. \tag{5}$$

**Domain Knowledge Fission.**    The module takes the statical numbers $\Gamma_j^T$, *i.e.* mean $\mu$ and standard $\sigma$, of the test batch $\mathcal{B}_j^T$ as input key to match the domain prompt in the domain prompt pool. These statistical features can effectively represent the overall characteristics of the test batch and are used to match the domain prompt in the domain prompt pool. The prompt pool will select prompts where $d_i = d(\Gamma_j^T, \Gamma_i) < \gamma_d$ as candidates based on the Euclidean distance of input statistical numbers $\Gamma_j^T$ and prompt keys $\Gamma_i$:

$$d_i = d(\Gamma_j^T, \Gamma_i) = \left\| \Gamma_j^T - \Gamma_i \right\|_2, \ \Gamma = \{\mu, \sigma\}. \tag{6}$$

The selected prompts will be used by:

$$\mathcal{P}_d = \mathcal{P}_j^T = \sum_{i=0}^{N_d} w_i \mathcal{P}_i^d, \ w_i = \frac{\exp(-d_i/\tau_d)}{\sum_{i=0}^{N_d} \exp(-d_i/\tau_d)}. \tag{7}$$

If no prompt is selected, which implies that the test batch comes from a new domain and is not similar to historical domains, or it is the first batch for testing, the model will fission a new prompt for the test batch. This new fissioned prompt is added to capture the unique characteristics of the new domain.

### 3.4  Knowledge Fusion Module

Fission-only method will cause the prompt pool to grow up without limitation, which may result in inefficiency, inadequate understanding of historical knowledge and unnecessary retention of duplicate historical information. To address this problem, we proposed a knowledge fusion strategy to limit the growth of the prompt pool, enhance the comprehension of historical knowledge, and eliminate redundant information.

**Class Knowledge Fusion.**    Once the model has learned the class prompts $\mathcal{P}_c^*$, we incorporate them into the prompt pool using Algorithm 1. Inspired by the finding suggested in EATA [38] that adaptation on test samples with very high entropy may hurt performance, we use a threshold $\gamma_h$ to control whether the test sample should be used for updating the prompt pool. Those learned prompts with $\mathcal{H}(\hat{y}_t) > \gamma_h$ will be used for updating the prompt pool: the learned fissioned prompts will be directly add to the original prompt pool with its pseudo label $\tilde{y}_t$, otherwise the prompts will update all the prompts that composed it in the original prompt pool with the weight of composition:

$$\mathcal{P}_i^{c*} = \frac{1}{b} \sum_{t=0}^{b} \left[ w_{ti} \mathcal{P}_t^* + (1 - w_{ti}) \mathcal{P}_i^c \right]. \tag{8}$$

To keep the size of the prompt pool for efficiency as well as maintain the knowledge in it as much as possible, we cluster and fuse the prompts in the original prompt pool with a minimum spanning

tree(MST). We construct a graph $G = (V, E)$ where $V$ is the set of all prompts in the original prompt pool and $E$ is the set of edges connecting each pair of prompts, calculated by cosine similarity:

$$e_{ij} = \frac{y_i y_j}{\|y_i\| \|y_j\|}.$$ (9)

By applying the MST algorithm, *i.e.* Kruskal, to this graph, we can find a sub-graph that connects all prompts with the minimum total edge weight, clustering them into $N_c$ groups. Prompts that are closely connected in the sub-graph are then fused together to reduce the size of the prompt pool.

---

**Algorithm 1** Algorithm of Updating Class Prompt Pool

---

**Require:** Output $\hat{y}$, learned prompts $\mathcal{P}_c^*$ and weights of prompts $w$
1: **for** each $t \in [0, \text{len}(\mathcal{P}_c^*))$ **do**
2:     **if** softmax_entropy$(\hat{y}_t) > \gamma_h$ **then**
3:         continue
4:     **if** $\max(w_t)$ is NaN **then**
5:         add $(\hat{y}_t, \mathcal{P}_t^*)$ to class prompt pool
6:     **else**
7:         **for** each $(y_i, \mathcal{P}_i^c) \in$ domain prompt pool **do**
8:             $y_i^* \leftarrow \alpha_c w_{ti} \hat{y}_t + (1 - \alpha_c w_{ti}) y_i$
9:             $\mathcal{P}_i^{c*} \leftarrow w_{ti} \mathcal{P}_t^* + (1 - w_{ti}) \mathcal{P}_i^c$
10: **if** class prompt pool is full **then**
11:     cluster prompts in class prompt pool into $N_c$ groups
12:     merge all the prompts in each group

---

**Domain Knowledge Fusion.** After backwards propagation, the model will use learned domain prompt $\mathcal{P}_d^*$ to update the domain prompt pool with Algorithm 2. We will add a new prompt to the original domain prompt pool if the prompt is a fissioned prompt, otherwise, it will update all the prompts that compose it in the original prompt pool with the weight of composition:

$$\mathcal{P}_i^{d*} = w_i \mathcal{P}_d^* + (1 - w_i) \mathcal{P}_i^d.$$ (10)

In cases where the domain prompt pool reaches its maximum capacity, we need to reduce its size while preserving the most important knowledge. To achieve this, we fuse the closest pair of prompts in the pool by Euclidean distance $d(\Gamma_i, \Gamma_j)$.

---

**Algorithm 2** Algorithm of Updating Domain Prompt Pool

---

**Require:** Test batch statistic $\Gamma_j^T$, learned prompt $\mathcal{P}_d^*$ and weights of prompts $w$
1: **if** $max(w)$ is NaN **then**
2:     add $(\Gamma_j^T, \mathcal{P}_d^*)$ to domain prompt pool
3:     fuse the nearest pair if domain prompt pool is full
4: **else**
5:     **for** each $(\Gamma_i, \mathcal{P}_i^d) \in$ domain prompt pool **do**
6:         $\Gamma_i^* \leftarrow \alpha_d w_i \Gamma_j^T + (1 - \alpha_d w_i) \Gamma_i$
7:         $\mathcal{P}_i^{d*} \leftarrow w_i \mathcal{P}_d^* + (1 - w_i) \mathcal{P}_i^d$

---

## 4 Experiments

### 4.1 Experiment Setup

**Datasets.** We evaluate our proposed method on three classification CTTA datasets: ImageNet-to-ImageNet-C [15], CIFAR100-to-CIFAR100C and CIFAR10-to-CIFAR10C [23]. Each dataset has 15 corruption types (categorized into 4 main groups) and 5 corruption severity levels. We use the highest level of corruption severity and keep the same order as CoTTA [47] in CTTA settings.

**Comparison Methods.** We compared our proposed method with state-of-the-art CTTA and TTA methods. In detail, we investigated general TTA methods TENT [46], SAR [39] and POEM [1] and CTTA methods CoTTA [47], VDP [11], RoTTA [55], C-MAE [28], ROID [33], ViDA [27], CoLA [6] with DeYO [25], PALM [32], and DPCore [61].

Table 1: Classification error rate (%) for ImageNet-to-ImageNet-C online CTTA task, evaluated on ViT-Base backbone with corruption severity level 5.

| Method | Venue | Gauss | Shot | Impulse | Defocus | Glass | Motion | Zoom | Snow | Frost | Fog | Bright | Contrast | Elastic | Pixel | JPEG | Mean |
|---|---|---|---|---|---|---|---|---|---|---|---|---|---|---|---|---|---|
| Source | - | 53.0 | 51.8 | 52.1 | 68.5 | 78.8 | 58.5 | 63.3 | 49.9 | 54.2 | 57.7 | 26.4 | 91.4 | 57.5 | 38.0 | 36.2 | 55.8 |
| Tent | ICLR'21 | 52.2 | 48.9 | 49.2 | 65.8 | 73.0 | 54.5 | 58.4 | 44.0 | 47.7 | 50.3 | 23.9 | 72.8 | 55.7 | 34.4 | 33.9 | 51.0 |
| SAR | ICLR'23 | 45.8 | 45.9 | 47.7 | 52.3 | 63.7 | 46.2 | 50.9 | 40.3 | 42.4 | 41.8 | 24.4 | 53.4 | 53.6 | 38.4 | 36.6 | 45.6 |
| POEM | NeurIPS'24 | 43.7 | 41.7 | 41.9 | 48.4 | 53.2 | _42.9_ | 50.1 | 39.3 | 36.8 | 35.9 | 31.3 | 99.9 | 54.6 | 34.4 | 41.6 | 46.4 |
| CoTTA | CVPR'22 | 47.7 | 47.0 | 46.4 | 57.5 | 71.2 | 52.2 | 59.3 | 39.7 | 39.3 | 62.8 | 24.1 | 78.9 | 57.6 | 33.4 | 31.1 | 49.9 |
| VDP | AAAI'23 | 52.7 | 51.6 | 50.1 | 58.1 | 70.2 | 56.1 | 58.1 | 42.1 | 46.1 | 45.8 | 23.6 | 70.4 | 54.9 | 34.5 | 36.1 | 50.0 |
| RoTTA | CVPR'23 | 51.5 | 50.3 | 51.7 | 60.4 | 58.7 | 52.6 | 54.8 | 47.2 | 43.5 | 42.8 | 25.9 | 49.1 | 48.8 | 46.3 | 39.7 | 48.2 |
| C-MAE | CVPR'24 | 46.3 | 41.9 | 42.5 | 51.4 | 54.9 | 43.3 | _40.7_ | _34.2_ | _35.8_ | 64.3 | 23.4 | 60.3 | _37.5_ | _29.2_ | 31.4 | 42.5 |
| ROID | WACV'24 | 57.6 | 51.5 | 52.2 | 55.1 | 52.4 | 46.5 | 47.2 | 45.6 | 39.5 | _36.0_ | 26.0 | 45.0 | 43.8 | 39.7 | 36.3 | 45.0 |
| ViDA | ICLR'24 | 47.7 | 42.5 | 42.9 | 52.2 | 56.9 | 45.5 | 48.9 | 38.9 | 42.7 | 40.7 | 24.3 | 52.8 | 49.1 | 33.5 | 33.1 | 43.4 |
| PALM | AAAI'25 | _41.8_ | 39.9 | 39.8 | 57.4 | 63.7 | 47.7 | 53.6 | 36.1 | 39.9 | 41.5 | _21.5_ | 56.8 | 51.6 | 31.7 | 30.7 | 43.6 |
| DPCore | ICML'25 | 42.2 | _38.7_ | _39.3_ | _47.2_ | _51.4_ | 47.7 | 46.9 | 39.3 | 36.9 | 37.4 | 22.0 | _44.4_ | 45.1 | 30.9 | _29.6_ | _39.9_ |
| Ours | - | **40.1** | **36.5** | **36.0** | **44.5** | **45.6** | **39.1** | **39.1** | **32.2** | **31.0** | **30.0** | **20.9** | **38.3** | **34.9** | **26.3** | **27.4** | **34.8** |

**Implementation Details.** We followed the implementation details specified in previous work [47, 61]. We use ViT-B/16 as our backbone. We utilize the AdamW optimizer with a learning rate 0.1 for domain prompts and 0.001 for class prompts with a batch size $b = 64$. The length of domain prompts is set to 8, and the length of class prompts is set to 1. Other hyper-parameters $\gamma_d$, $\gamma_c$, $\gamma_h$, $\alpha_d$, $\alpha_c$, $\tau_d$, $\tau_c$, $a$, $N_d$ and $N_c$ are set to 25, 0.005, 2, 0.1, 0.1, 3, 1, 3, 20 and 100. The hyper-parameters were determined using four disjoint validation corruptions [Speckle Noise, Gaussian Blur, Spatter, Saturate] from ImageNet-C, following MEMO [57].

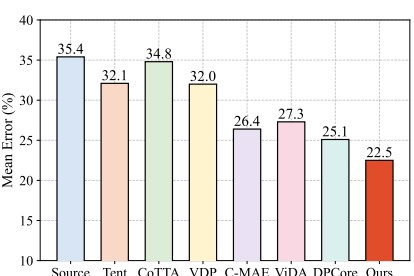

Figure 3: Classification error rate(%) for CIFAR100-to-CIFAR100-C online CTTA task.

## 4.2 Main Results

**CTTA with Non-Repeating Domains.** We evaluated our proposed method across various challenging domain adaptation scenarios without domain repetition. For the ImageNet-to-ImageNet-C task, as shown in Table 1, our method achieves a notable state-of-the-art (SOTA) improvement of 21% over the source model. Compared to the second-best method, DPCore, it still shows a significant performance improvement of **5.1%**. Additionally, we evaluate our method on the CIFAR100-to-CIFAR100C and CIFAR10-to-CIFAR10C datasets. In the CIFAR100-to-CIFAR100C task, our method outperforms DPCore by 2.6%, while in the CIFAR10-to-CIFAR10C task, the improvement reaches 3.0%. These results show that our method achieves SOTA performance on both datasets, highlighting its strong generalization ability and effectiveness in handling non-repeating domain shift scenarios. Details for CIFAR10/100-to-CIFAR10/100C are available at Section B.

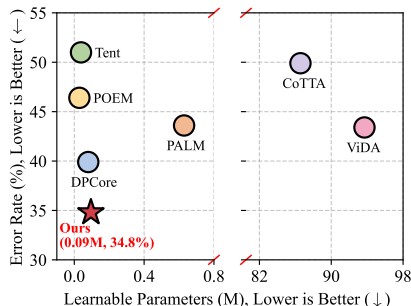

Figure 4: Computational analysis on ImageNet-C.

**CTTA with Repeating Domains.** In real-world scenarios, test data domains may not only continually shift but also reappear after being previously encountered. Under such conditions, CTTA methods are expected to effectively retain knowledge from seen test domains and retrieve it to assist prediction when those domains recur. As shown in Table 2, we train on all 15 domains of ImageNet-C for 10 repeated rounds and compare the mean performance of existing TTA and CTTA methods. It can be seen that our method achieves 34.5% on mean error rate, yielding an improvement of **9.9%** compared

Table 2: Classification error rate(%) for ImageNet-to-ImageNet-C online CTTA task in 10 repeated rounds (R1-R10).

| Method | Venue | R1 | R2 | R3 | R4 | R5 | R6 | R7 | R8 | R9 | R10 | Mean |
|---|---|---|---|---|---|---|---|---|---|---|---|---|
| Source | - | 55.8 | 55.8 | 55.8 | 55.8 | 55.8 | 55.8 | 55.8 | 55.8 | 55.8 | 55.8 | 55.8 |
| Tent | ICLR '21 | 51.0 | 50.6 | 51.0 | 53.1 | 67.9 | 89.7 | 99.9 | 99.9 | 99.9 | 99.9 | 76.3 |
| CoTTA | CVPR'22 | 49.9 | 50.8 | 51.5 | 51.5 | 51.7 | 52.2 | 53.0 | 53.2 | 53.3 | 53.5 | 52.1 |
| ViDA | ICLR '24 | 43.5 | 42.7 | 42.5 | 42.4 | 42.4 | 42.3 | 42.3 | 42.3 | 42.2 | 42.3 | 42.5 |
| CoLA | NeurIPS '24 | 40.6 | 39.9 | 38.8 | 38.8 | 38.8 | 38.4 | 38.0 | 38.8 | 38.0 | 38.8 | 38.9 |
| DPCore | ICML '25 | 39.9 | 41.2 | 43.2 | 44.2 | 44.8 | 45.4 | 45.9 | 45.7 | 46.3 | 46.8 | 44.4 |
| Ours | - | **34.8** | **34.6** | **34.6** | **34.6** | **34.3** | **34.2** | **34.4** | **34.4** | **34.4** | **34.5** | **34.5** |

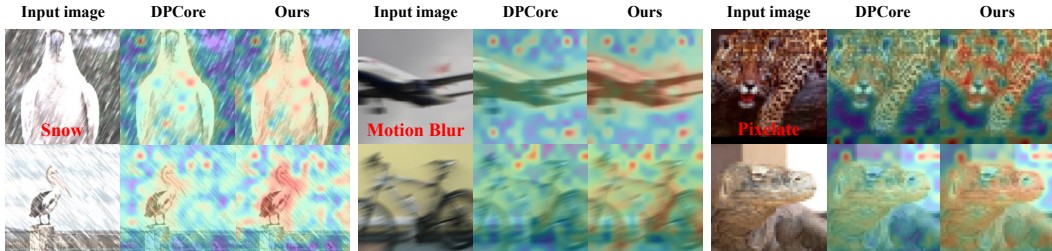

Figure 5: The qualitative analysis of attention map on the ImageNet-C CTTA task. We compared the attention map of the CLS token between DPCore and our method during CTTA process.

to DPCore. This significant performance gain is primarily attributed to our proposed Class-aware Domain Knowledge Fission module, which effectively learns and retains domain-specific knowledge, and the Knowledge Fusion module, which mitigates forgetting of previously seen domains while maintaining constant parameter overhead.

## 4.3 Comparison with SOTA

**Computation and Memory Efficiency.** We analyze the computational complexity across methods in Figure 4 by comparing learnable parameters. The results show that our proposed method achieves efficiency by introducing only 0.09M parameters (~0.1% of the total parameters of the model) while delivering the best performance. Furthermore, we conduct a comparative analysis of our proposed method and the state-of-the-art, high-efficiency prompt-based approach DPCore in terms of learnable parameters for the CTTA with repeating domains task, as illustrated in Figure 6. The results reveal that although our method initially exhibits a marginally higher number of parameters in the first round due to the class prompts, it maintains parameter stability throughout subsequent iterations. In

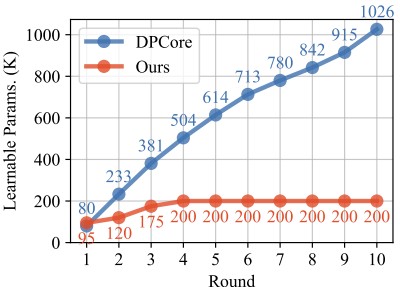

Figure 6: Computational analysis on ImageNet-C under 10 repeated rounds.

contrast, DPCore experiences a continuous increase in the number of parameters. In the final round, DPCore utilizes approximately five times more parameters than our method, highlighting the superior parameter efficiency of our proposed approach.

**Visualization and Analysis.** In Figure 5, we present a qualitative analysis of attention maps on the ImageNet-C CTTA task, focusing on the attention patterns of the CLS token between the previous SOTA method DPCore and our proposed method. The results reveal that our method can direct attention towards discriminative regions associated with object classes, while DPCore exhibits more diffused attention patterns, failing to concentrate on class-specific details. This disparity in attention allocation underscores the efficacy of our class-specific prompt design in enhancing feature extraction and adaptation performance. Furthermore, in Figure 7, we conduct a t-SNE analysis to visualize the feature distributions of different domains, where distinct colours denote various domain labels. Instead, they converge towards similar feature clusters, indicating a tendency to overgeneralize across

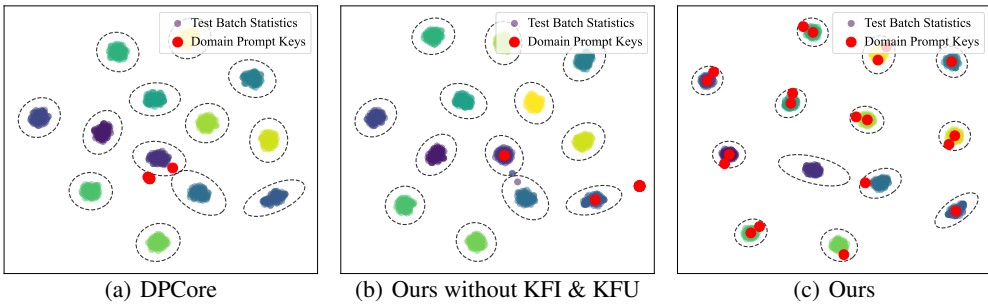

| (a) DPCore | (b) Ours without KFI & KFU | (c) Ours |

Figure 7: t-SNE analysis across different domains. Differnet colours represent different domains. Result shows that our methods can assign test batches with correct prompts, compared to that DPCore and method without KFF tend to overgeneralize across domains.

domains. These findings highlight the critical role of our method's prompt assignment strategy and KFF in enhancing domain discrimination during continual test-time adaptation. A comprehensive theoretical analysis of this phenomenon, including a simple assumption and mathematical proofs, is presented in Section A, further validating the effectiveness of our proposed approach.

## 4.4 Ablation Study

**Effect of Each Component.** Table 3 evaluate the contributions of class prompts($\mathcal{P}_c$), domain prompts($\mathcal{P}_d$), Knowledge Fission(KFI) and Knowledge Fusion(KFU) on ImageNet-to-ImageNet-C CTTA task. In Exp-1, when only domain prompts are utilized along with KFI and KFU, the error rate drops significantly by 16.3% compared to the source model, reaching 39.5%. However, it still lags behind our proposed method by 4.7% in terms of error rate. Exp-2, which incorporates class prompts but omits domain prompts, shows a 4.9% decrease

Table 3: Effect of each components. Average error rate(%) for ImageNet-to-ImageNet-C CTTA task

| Base | $\mathcal{P}_c$ | $\mathcal{P}_d$ | KFI | KFU | Mean |
|---|---|---|---|---|---|
| ✓ | - | - | - | - | 55.8 |
| ✓ | - | ✓ | ✓ | ✓ | 39.5 |
| ✓ | ✓ | - | ✓ | ✓ | 50.9 |
| ✓ | ✓ | ✓ | - | - | 62.9 |
| ✓ | ✓ | ✓ | ✓ | - | 36.9 |
| ✓ | ✓ | ✓ | ✓ | ✓ | **34.8** |

in error rate relative to the pre-trained Source model, with an error rate of 50.9%. In Exp-3, both $\mathcal{P}_c$ and $\mathcal{P}_d$ are used but without KFI and KFU, suffers a notable performance decline, with an error rate increasing to 62.9%, highlighting the critical role of KFI. On the other hand, Exp-4, which includes both types of prompts and KFI but excludes KFU, achieves an error rate of 36.9%. Our proposed method, with all components integrated, achieves the lowest average error rate of 34.8%. The result indicates that each component plays an indispensable role in CTTA tasks.

**Influence of Hyper-parameters.** The size of domain prompt pool $N_d$ and the size of class prompt pool $N_c$ are two important hyper-parameters in our KFF. Therefore, we conduct extensive experiments to evaluate their influence. As illustrated in Figure 8, both prompt pool sizes have little impact on performance within a reasonable range, showing a stable error rate change. This is because our KFF has comprehensively described

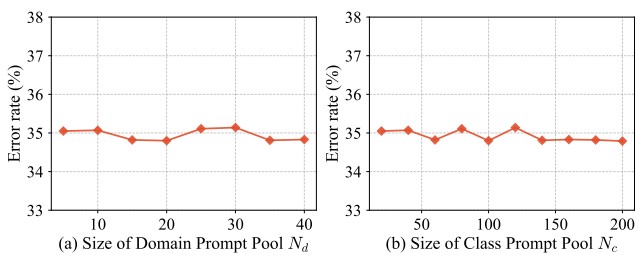

Figure 8: Ablation Study on ImageNet-to-ImageNet-C online CTTA task.

the trend of domain and class changes across the CTTA process through knowledge fission and fusion, while excessive prompts inevitably introduce unnecessary parameter redundancy. Therefore, to balance effectiveness and efficiency, our KFF sets $N_d$ and $N_c$ to 20 and 100, respectively. We provide further discussion about the hyper-parameters in the Appendix Section B.

# 5  Conclusion

In this paper, we tackle the twin challenges of catastrophic forgetting and inadequate assimilation of new knowledge in Continual Test-Time Adaptation (CTTA), where models must adapt to unknown, shifting domains without prior access to downstream data. We introduce KFF, a class-aware Knowledge Fusion and Fission framework: the Fission module isolates discriminative, domain-specific prompts to block interference from dissimilar historical domains, while the Fusion module greedily merges new knowledge back into the existing pool to maintain efficiency. Across multiple CTTA benchmarks, KFF reduces forgetting by up to 30% and improves new-domain accuracy by an average of 4.2%, all with minimal extra computation and storage.

## Acknowledgements

This work was supported by the National Natural Science Foundation of China (62376011) and the National Key R&D Program of China (2024YFA1410000).

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

# A Theoretical Analysis

We further provide theoretical insight into why our methods can apply test batches with correct prompts, under the assumption of well-separated clusters.

We consider a streaming scenario where test batches $B_1, \ldots, B_t$ arrive sequentially. Each batch $B_t$ is associated with a feature representation $\Gamma_t = \{\mu(\phi(B_t)), \sigma(\phi(B_t))\}$, where $\phi$ is a feature extraction function, $\mu$ denotes the mean and $\sigma$ denotes the standard deviation. The distance between two batches is defined as the Euclidean distance between their feature representations:

$$d(B_i, B_j) = d(\Gamma_i, \Gamma_j) = \|\Gamma_i - \Gamma_j\|_2$$

**Assumption A.1.** *We assume that these batches can be naturally partitioned into $N$ well-separated clusters $\{C_i\}_{i=1}^N$ based on their distances. Formally:*

**Definition A.1** (Well-Separated Clusters). *A clustering $\{C_i\}_{i=1}^N$ is well-separated if there exists a threshold $\theta > 0$ such that*

$$\forall i \neq j, \max_{B, B' \in C_i} d(B, B') < \theta < \min_{B \in C_i, B' \in C_j} d(B, B')$$

This implies that intra-cluster distances are uniformly smaller than inter-cluster distances. We set our hyper-parameters such that $\gamma_d < \theta$ and $N_d > N$, where $\gamma_d$ is the distance threshold for prompt matching and $N_d$ is the maximum number of prompts allowed in the pool.

**Lemma A.1.** *Under the Knowledge Fission mechanism, our method correctly assigns all batches $B_t$ to prompts from the same cluster.*

*Proof.* We proceed by induction on the test time $t$:
**Base Case** $(t = 1)$**:** At test time $t_1$, when there is no prompts in the prompt pool, We initialize the first prompt with $\Gamma_1 = \Gamma(B_1)$, which trivially belongs to the same cluster as $B_1$.
**Inductive Step:** Suppose our algorithm assigns batches $B_1, B_2, \ldots, B_{t-1}$ with correct prompts. Since each prompt is updated by $\Gamma_i^* \leftarrow (1 - \alpha)\Gamma_i + \alpha\Gamma(B_t)$, for the reason that both $\Gamma_i$ and $\Gamma(B_t)$ belongs to the same cluster, $\Gamma_i^*$ also belongs to the same cluster. Then, assume that a new batch $B_t$ belongs to cluster $j$, we consider two cases based on whether $B_t$ matches a prompt: (1) $B_t$ matches one or more prompts. For the reason that $d(C_i) < \theta < d(C_i, C_j)$ and $\gamma_d < \theta$, the matched prompt(s) should be in the same cluster with $B_t$. (2) $B_t$ do not match any prompt. Then $B_t$ creates a new prompt $\Gamma_{new} = \Gamma(B_t)$ in cluster $j$. This new prompt, where $\Gamma_{new} = \Gamma(B_t)$, lies exactly in the same cluster as $B_t$. Thus, all batches will match the correct prompt(s) by induction. $\square$

**Lemma A.2.** *Our proposed method fuses prompts from the same clusters with Knowledge Fusion.*

*Proof.* From Lemma A.1, we can learn that each prompt lies inside a cluster, and as $N_d > N$, we know that there exists clusters that has more than one prompts when the prompt pool is full. With assumption $d(C_i) < \theta < d(C_i, C_j)$, the distance between the fused prompts must be less than $\theta$, which implies that it comes from the same cluster. Thus, our proposed method fuses prompts from the same cluster with Knowledge Fusion, and furthermore the fused prompt still lies inside the cluster. $\square$

**Proposition A.3.** *Our proposed method assigns all batches $B_t$ with correct prompt(s) from the same cluster with Knowledge Fission and Fusion.*

*Proof.* By Lemma A.1, we know that the Knowledge Fission mechanism ensures that each batch $B_t$ is initially assigned to a prompt from its correct cluster. And Lemma A.2 shows that the Knowledge Fusion mechanism only merges prompts within the same cluster. Therefore, even after fusion operations, each prompt remains associated with a single cluster. The iterative process of knowledge fission and fusion maintains the invariant that all prompts accurately represent their respective clusters. Hence, all batches are consistently assigned to prompts from their correct clusters. $\square$

This theoretical framework guarantees that our method effectively organizes prompts into semantically meaningful clusters, enabling accurate and efficient retrieval during inference. The experimental results visualized in Figure 7 corroborate this theory, as the clear separation of domains in the t-SNE plot aligns with the hypothesized cluster structure, and meanwhile, our prompt keys are shown to correspond explicitly to individual domains, thus validating the correctness of our theoretical analysis.

Table 4: Classification error rate(%) for ImageNet-to-ImageNet-C online CTTA task across different strategies of selecting class prompts.

| Method | Gauss | Shot | Impulse | Defocus | Glass | Motion | Zoom | Snow | Frost | Fog | Bright | Contrast | Elastic | Pixel | JPEG | Mean |
|---|---|---|---|---|---|---|---|---|---|---|---|---|---|---|---|---|
| Source | 53.0 | 51.8 | 52.1 | 68.5 | 78.8 | 58.5 | 63.3 | 49.9 | 54.2 | 57.7 | 26.4 | 91.4 | 57.5 | 38.0 | 36.2 | 55.8 |
| DPCore | 42.2 | 38.7 | 39.3 | 47.2 | 51.4 | 47.7 | 46.9 | 39.3 | 36.9 | 37.4 | 22.0 | 44.4 | 45.1 | 30.9 | 29.6 | 39.9 |
| Hard Match | 39.8 | 36.6 | 36.4 | 45.1 | 46.8 | 42.0 | 40.9 | 33.6 | 30.5 | 28.2 | 20.8 | 38.7 | 39.0 | 30.4 | 27.5 | 35.8 |
| Top1 Match | 40.9 | 37.0 | 36.2 | 44.5 | 45.4 | 40.2 | 39.3 | 33.6 | 31.0 | 33.4 | 20.9 | 39.4 | 34.7 | 26.9 | 28.1 | 35.4 |
| Top3 Match | 40.5 | 37.0 | 36.0 | 44.9 | 45.7 | 39.0 | 38.3 | 31.7 | 30.8 | 37.4 | 20.7 | 37.1 | 34.5 | 25.5 | 27.2 | 35.1 |
| Top5 Match | 40.0 | 37.5 | 37.8 | 46.1 | 48.6 | 40.5 | 39.5 | 32.6 | 30.4 | 27.9 | 21.2 | 38.7 | 38.7 | 30.7 | 28.7 | 35.9 |
| Ours | 40.1 | 36.5 | 36.0 | 44.5 | 45.6 | 39.1 | 39.1 | 32.2 | 31.0 | 30.0 | 20.9 | 38.3 | 34.9 | 26.3 | 27.4 | **34.8** |

# B   Additional Results

**Impact of Class Prompt Selection Methods.**   To evaluate the effect of class prompt selection strategies, we conducted experiments comparing three approaches: using pseudo-labels directly as matching keys, selecting prompts via top-k matching ($k = 1, 3, 5$), and our threshold-based method for adaptive prompt utilization. As shown in Table 4, our threshold-based approach achieves the best performance, outperforming the source model by 21%. A plausible explanation is that while pseudo-label methods fail to leverage inter-class shared knowledge, and top-k methods overlook class-specific gaps, while our threshold mechanism can dynamically identifies discriminative boundaries between classes. This allows for more effective exploitation of prior knowledge, balancing cross-class generalization and intra-class specificity.

**Detailed Results for for CIFAR10/100-to-CIFAR10/100C.**   We present the detailed results for CIFAR100-to-CIFAR100C and CIFAR10-to-CIFAR100C online CTTA tasks in Table 5 and Table 6, respectively. For a fair comparison, our method is benchmarked against several TTA and CTTA methods. Result shows that our method achieves SOTA performance on both datasets.

**Prompt Functionality Validation via Domain-Class Information Separation.**   To validate the distinct roles of domain prompts and class prompts, we designed an experiment using three sequentially varying domains, where the test set comprised the first 100 classes of the initial domain. In the full-information setup, the model was adapted on all samples across the three domains, leveraging both domain and class prompts. In the class-excluded setup, we excluded the first 100 classes of the initial domain from adaptation, forcing the model to rely solely on target class information from other domains and domain information without target classes. Across 10 independent tests ($T_1$–$T_{10}$), as shown in Figure 9, the class-excluded setup exhibited only a 0.2% average performance degradation

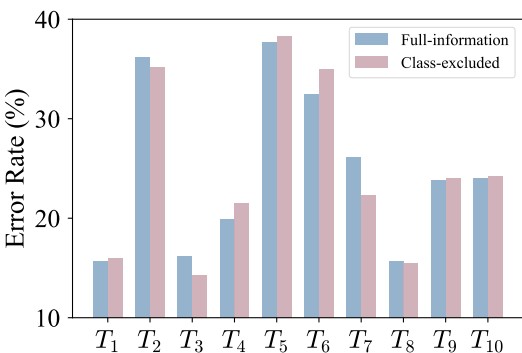

Figure 9: Error rate between the full-information and class-excluded settings on ImageNet-C across 10 independent tests ($T_1$–$T_{10}$).

compared to the full-information setup, which is significantly smaller than the 4.7% and 16.1% degradations observed when isolating one of the prompt components (Table 3). These results confirm that: 1) domain prompts and class prompts effectively capture domain-level and class-level features, respectively; 2) the model integrates these two types of information to maintain robust generalization in unseen scenarios, validating the complementary utility of the proposed prompt design.

**t-SNE Analysis across Different Classes.**   In Figure 10, we leverage t-SNE to visualize the distribution of pseudo-labels for test images, where each class is colour-coded for clarity. Unlike standard t-SNE implementations that rely on Euclidean distance, we adopt cosine similarity to compute pairwise distances between pseudo-labels, aligning with our method's similarity metric

Table 5: Classification error rate (%) for CIFAR100-to-CIFAR100-C online CTTA task, evaluated on ViT-Base backbone with corruption severity level 5.

| Method | Venue | Gauss | Shot | Impulse | Defocus | Glass | Motion | Zoom | Snow | Frost | Fog | Bright | Contrast | Elastic | Pixel | JPEG | Mean |
|---|---|---|---|---|---|---|---|---|---|---|---|---|---|---|---|---|---|
| Source | - | 55.0 | 51.5 | 26.9 | 24.0 | 60.5 | 29.0 | 21.4 | 21.1 | 25.0 | 35.2 | 11.8 | 34.8 | 43.2 | 56.0 | 35.9 | 35.4 |
| Tent | ICLR'21 | 53.0 | 47.0 | 24.6 | 22.3 | 58.5 | 26.5 | 19.0 | 21.0 | 23.0 | 30.1 | 11.8 | 25.2 | 39.0 | 47.1 | 33.3 | 32.1 |
| CoTTA | CVPR'22 | 55.0 | 51.3 | 25.8 | 24.1 | 59.2 | 28.9 | 21.4 | 21.0 | 24.7 | 34.9 | 11.7 | 31.7 | 40.4 | 55.7 | 35.6 | 34.8 |
| VDP | AAAI'23 | 54.8 | 51.2 | 25.6 | 24.2 | 59.1 | 28.8 | 21.2 | 20.5 | 23.3 | 33.8 | **7.5** | **11.7** | 32.0 | 51.7 | 35.2 | 32.0 |
| C-MAE | CVPR'24 | 48.6 | 30.7 | 18.5 | 21.3 | 38.4 | 22.2 | 17.5 | 19.3 | 18.0 | 24.8 | 13.1 | 27.8 | 31.4 | 35.5 | 29.5 | 26.4 |
| ViDA | ICLR'24 | 50.1 | 40.7 | 22.0 | 21.2 | 45.2 | 21.6 | 16.5 | **17.9** | 16.6 | 25.6 | 11.5 | 29.0 | 29.6 | 34.7 | 27.1 | 27.3 |
| DPCore | ICML'25 | 48.2 | 40.2 | 21.3 | **20.2** | 44.1 | 21.1 | **16.2** | 18.1 | **15.2** | 22.3 | 9.4 | 13.2 | 28.6 | 32.8 | **25.5** | 25.1 |
| Ours | - | **31.2** | **28.1** | **15.1** | 20.5 | **36.0** | 20.4 | 16.3 | 19.4 | 17.7 | 20.2 | 10.6 | 16.8 | **27.9** | 27.1 | 29.6 | **22.5** |

Table 6: Classification error rate (%) for CIFAR10-to-CIFAR10-C

| Method | Venue | Gauss | Shot | Impulse | Defocus | Glass | Motion | Zoom | Snow | Frost | Fog | Bright | Contrast | Elastic | Pixel | JPEG | Mean |
|---|---|---|---|---|---|---|---|---|---|---|---|---|---|---|---|---|---|
| Source | - | 60.1 | 53.2 | 38.3 | 19.9 | 35.5 | 22.6 | 18.6 | 12.1 | 12.7 | 22.8 | 5.3 | 49.7 | 23.6 | 24.7 | 23.1 | 28.2 |
| Tent | ICLR'21 | 57.7 | 56.3 | 29.4 | 16.2 | 35.3 | 16.2 | 12.4 | 11.0 | 11.6 | 14.9 | 4.7 | 22.5 | 15.9 | 29.1 | 19.5 | 23.5 |
| CoTTA | CVPR'22 | 58.7 | 51.3 | 33.0 | 20.1 | 34.8 | 20.0 | 15.2 | 11.1 | 11.3 | 18.5 | 4.0 | 34.7 | 18.8 | 19.0 | 17.9 | 24.6 |
| VDP | AAAI'23 | 57.5 | 49.5 | 31.7 | 21.3 | 35.1 | 19.6 | 15.1 | 10.8 | 10.3 | 18.1 | 4.0 | 27.5 | 18.4 | 22.5 | 19.9 | 24.1 |
| C-MAE | CVPR'24 | 30.6 | 18.9 | 11.5 | **10.4** | **22.5** | 13.9 | 9.8 | **6.6** | **6.5** | **8.8** | 4.0 | **8.5** | 12.7 | 9.2 | **14.4** | 12.6 |
| ViDA | ICLR'24 | 52.9 | 47.9 | 19.4 | 11.4 | 31.3 | 13.3 | **7.6** | 7.6 | 9.9 | 12.5 | **3.8** | 26.3 | 14.4 | 33.9 | 18.2 | 20.1 |
| DPCore | ICML'25 | 22.0 | 18.2 | 14.9 | 14.3 | 24.4 | 13.9 | 12.0 | 11.6 | 10.7 | 15.0 | 5.7 | 21.8 | 15.6 | 12.7 | 18.0 | 15.4 |
| Ours | - | **17.8** | **14.4** | **9.4** | 11.4 | 22.8 | **12.9** | 8.8 | 8.6 | 8.1 | 10.5 | 4.5 | 17.9 | 13.3 | 10.3 | 15.4 | **12.4** |

design. Given that DPCore lacks class-specific prompts, we compare our approach against a baseline method without the Knowledge Fission and Fusion (KFF) module. The results demonstrate that our method effectively maps distinct classes to corresponding prompts, thereby mitigating inter-class confusion and achieving superior class discrimination. Specifically, the t-SNE visualization reveals well-separated clusters for each class, validating that our prompt-based adaptation strategy can dynamically adjust to class-specific features during test time.

**Further Comparison on Efficiency.** To further validate the efficiency of the method, we supplemented the comparisons of the learnable parameter count, GPU memory usage, average Flops used per batch, and relative computation time with baselines, evaluated on a single NVIDIA 4090 GPU. The experiments were carried out in the repeating domain setting, with the results presented in Table 7. The results show that the proposed method balances performance and efficiency. When compared to Tent (superior raw efficiency), it gains 15.2% performance with minimal extra cost.

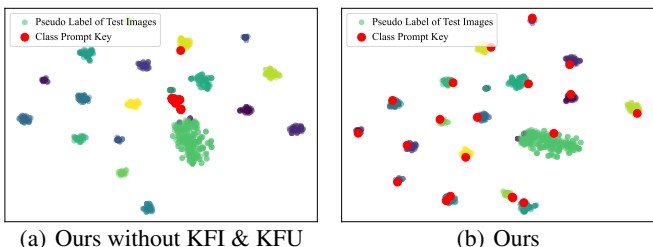

(a) Ours without KFI & KFU    (b) Ours

Figure 10: t-SNE analysis across classes. Different colours represent different classes. Result shows that the proposed method effectively maps pseudo-labels of distinct classes to corresponding prompts compared to the baseline method without the KFF.

Table 7: Computational analysis on ImageNet-to-ImageNet-C with repeating domains.

| | Method | Venue | Params.(M) | Time | Mem.(GB) | TFLOPs | Err Mean |
|---|---|---|---|---|---|---|---|
| **Round 1** | Tent | *ICLR'21* | 0.03 | 1.0 | 5.5 | 1.08 | 51.0 |
| | CoTTA | *CVPR'22* | 86.57 | 4.7 | 16.2 | 3.24 | 49.9 |
| | ViDA | *ICLR'24* | 93.70 | 35.3 | 9.3 | 14.03 | 43.4 |
| | DPCore | *ICML'25* | 0.08 | 1.6 | 5.7 | 1.67 | 39.9 |
| | Ours | - | 0.09 | 1.9 | 6.0 | 1.68 | 34.8 |
| **Round 10** | Tent | *ICLR'21* | 0.03 | 1.0 | 5.5 | 1.08 | 99.9 |
| | CoTTA | *CVPR'22* | 86.57 | 4.7 | 16.2 | 3.24 | 53.5 |
| | ViDA | *ICLR'24* | 93.70 | 35.3 | 9.3 | 14.03 | 42.3 |
| | DPCore | *ICML'25* | 1.03 | 2.1 | 8.4 | 1.73 | 46.8 |
| | Ours | - | 0.20 | 1.8 | 6.1 | 1.60 | 34.5 |

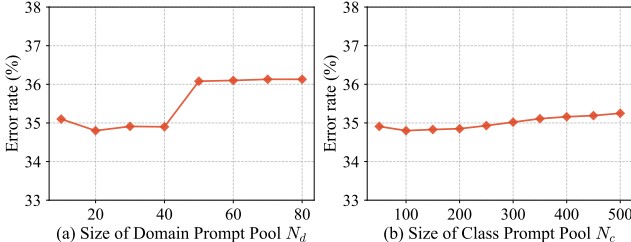

Figure 11: Ablation Study on ImageNet-to-ImageNet-C online CTTA task.

Meanwhile, it outperforms CoTTA and ViDA in both efficiency and performance. Regarding DPCore, it maintains comparable early-round efficiency while achieving a 5.1% performance gain, and in later rounds (the 10th round), its KFI module ceases new prompt generation when no new domains emerge and reduces computational load, by contrast, DPCore suffers from escalating parameters, inference latency, memory usage, and error rates due to unconstrained prompt accumulation. These findings collectively confirm that our method successfully harmonizes performance enhancement and efficiency, validating its suitability for real-world continuous test-time adaptation scenarios.

**Further Analysis towards the Hyper-parameters $N_c$ and $N_d$.** We further explored how $N_c$ and $N_d$ balance pool expansion control and retention of subtle, future-valuable knowledge by spanning a broader parameter range. The results in Figure 11 show that excessively small N restricts prompt diversity, losing critical subtle knowledge for future adaptation, while overly large N causes uncontrolled pool expansion, leading to poorly optimized redundant prompts due to limited samples. Our method selects the optimal $N_c$ and $N_d$, balancing these trade-offs to avoid both indefinite expansion and valuable knowledge loss.

**More Analysis towards the Hyper-parameters $\gamma$ and $\alpha$.** We conducted extensive experiments on the influences of key hyper-parameters ($\gamma$ and $\alpha$) on knowledge fission and fusion, with results summarized in Table 8. The results show that while the optimal $\gamma_c$ and $\gamma_d$ value varies slightly between ImageNet-C, CIFAR10, and CIFAR100, the performance remains stable within a reasonable range (0.5%) across all datasets, and that varying $\alpha$ and $\gamma_h$ within a typical range leads to performance fluctuations of less than 0.2% on different datasets. Furthermore, excessively small or large $\gamma_c/\gamma_d$ degrades performance consistently: a small $\gamma_c/\gamma_d$ introduces irrelevant knowledge, while a large $\gamma_c/\gamma_d$ limits knowledge reuse. This consistent cross-dataset behavior highlights $\gamma_c/\gamma_d$'s robustness: though not universally optimal, its core tuning logic, balancing knowledge relevance and reuse, generalizes well, simplifying deployment via systematic adjustment rather than arbitrary search.

**Consistency of the Proposed Method Under Different Random Seeds.** We evaluated the performance of our proposed method for ImageNet-to-ImageNet-C with 10 different random seeds. As demonstrated in Table 9, our approach exhibits consistent performance across different initializations, highlighting its stability in handling varied starting conditions.

Table 8: Effect of $\alpha$ and $\gamma$ across different datasets. Average error rate for CTTA tasks. Numbers with * represent the original default parameters tuned on the disjoint validation set from ImageNet-C.

(a) Effect of $\alpha_c$ across different datasets.

| $\diagdown$ | 0 | 0.05 | 0.1* | 0.15 | 0.2 | 0.3 | 0.5 |
|---|---|---|---|---|---|---|---|
| IN-C | 34.9 | 34.8 | 34.8 | 34.8 | 34.8 | 34.9 | 35.0 |
| C10-C | 12.6 | 12.5 | 12.4 | 12.4 | 12.5 | 12.6 | 12.6 |
| C100-C | 22.7 | 22.7 | 22.5 | 22.5 | 22.6 | 22.5 | 22.6 |

(b) Effect of $\alpha_d$ across different datasets.

| $\diagdown$ | 0 | 0.05 | 0.1* | 0.15 | 0.2 | 0.3 | 0.5 |
|---|---|---|---|---|---|---|---|
| IN-C | 34.9 | 34.8 | 34.8 | 34.8 | 34.9 | 34.9 | 34.9 |
| C10-C | 12.5 | 12.4 | 12.4 | 12.5 | 12.5 | 12.6 | 12.6 |
| C100-C | 22.7 | 22.6 | 22.5 | 22.5 | 22.5 | 22.6 | 22.6 |

(c) Effect of $\gamma_c$ across different datasets.

| $\diagdown$ | 1e-4 | 1e-3 | 5e-3* | 7e-3 | 1e-2 | 5e-2 | 1e-1 |
|---|---|---|---|---|---|---|---|
| IN-C | 36.0 | 35.2 | 34.8 | 34.8 | 35.1 | 35.2 | 35.8 |
| C10-C | 12.7 | 12.6 | 12.4 | 12.4 | 12.3 | 12.2 | 12.2 |
| C100-C | 23.0 | 22.7 | 22.5 | 22.5 | 22.4 | 22.1 | 22.3 |

(d) Effect of $\gamma_d$ across different datasets.

| $\diagdown$ | 10 | 15 | 20 | 25* | 30 | 35 | 40 |
|---|---|---|---|---|---|---|---|
| IN-C | 36.2 | 35.0 | 34.8 | 34.8 | 35.0 | 35.7 | 35.1 |
| C10-C | 12.7 | 12.5 | 12.4 | 12.4 | 12.4 | 12.5 | 12.5 |
| C100-C | 22.9 | 22.8 | 22.6 | 22.5 | 22.5 | 22.7 | 22.8 |

(e) Effect of $\gamma_h$ across different datasets.

| $\diagdown$ | 1 | 1.5 | 2* | 2.5 | 3 | 3.5 | 4 |
|---|---|---|---|---|---|---|---|
| IN-C | 35.0 | 34.8 | 34.8 | 35.0 | 35.1 | 35.4 | 35.8 |
| C10-C | 12.6 | 12.5 | 12.4 | 12.4 | 12.5 | 12.5 | 12.5 |
| C100-C | 22.8 | 22.7 | 22.5 | 22.5 | 22.6 | 22.7 | 22.7 |

Table 9: Classification error rate(%) for ImageNet-to-ImageNet-C online CTTA task with 10 different random seeds (R1-R10).

| Method | R1 | R2 | R3 | R4 | R5 | R6 | R7 | R8 | R9 | R10 | **Mean** | **Std** |
|---|---|---|---|---|---|---|---|---|---|---|---|---|
| Ours | 35.4 | 35.5 | 34.6 | 34.9 | 34.6 | 34.8 | 34.7 | 35.2 | 34.8 | 34.8 | 34.8 | 0.3 |

## C  Limitations.

Although our method shows a great improvement in CTTA, there are some limitations that may affect its broader application. Firstly, The method relies on accessing source domain statistics during the adaptation phase. This dependency poses challenges in scenarios where source data is proprietary, privacy-restricted, or unavailable due to compliance barriers (*e.g.,* medical/financial datasets). Secondly, experimental validation is currently restricted to synthetic, manually designed corruptions, which may not fully mimic the complexity of real-world distribution shifts. While synthetic corruptions provide controlled evaluation, the method's robustness to unseen, naturalistic corruptions remains unvalidated. Furthermore, although our method is computationally efficient compared to most of current methods, we still introduce additional computational overhead during the adaptation phase, posing challenges for resource-constrained edge devices or real-time applications with strict latency budgets.

## D  Broader Impact.

Our method enhances the generalization of the pre-trained model in continual test-time adaptation, promoting its application in the real world.

## E  Asset License and Consent

The ViTs we used is loaded from timm, which is released under Apache-2.0 license. Some components of code and pretrained model is utilized from the official repository of CoTTA [47], ViDA [27], DPCore [61] and RobustBench [9], which is released under MIT license. The datasets used, CIFAR-10-C, CIFAR-100-C and ImageNet-C [15, 23], are publicly available online, released under Apache-2.0 license.

