# OpenReview forum: "Class-aware Domain Knowledge Fusion and Fission for Continual Test-Time Adaptation"
_NeurIPS.cc/2025/Conference — NeurIPS 2025 poster_

### Official Review · Reviewer_8gyU · 2025-06-26

**Clarity:** 3
**Significance:** 2
**Originality:** 3
**Rating:** 4
**Confidence:** 4

**Summary:**

This paper proposes a visual prompt tuning-based method for Continual Test-Time Adaptation (CTTA). Specifically, the method introduces a class prompt pool and a domain prompt pool. A fusion-and-fission mechanism is designed to update these pools by expanding prompts for new classes and domains, and refining existing prompts for closely related classes and domains. Experiments are conducted on commonly used benchmark datasets in the CTTA literature to evaluate the proposed method.

**Questions:**

1. Could the authors elaborate on the conceptual and technical contributions of the proposed method in comparison to prior visual prompt tuning-based CTTA methods, such as DPCore?
2. As mentioned in Weakness 2, I recommend conducting a more comprehensive efficiency analysis beyond the limited parameter count comparison.
3. How is the data stream constructed? Is it sampled in an independent and identically distributed manner, similar to regular training? Have the evaluations considered temporally correlated data streams, as in baselines like RoTTA [47] and SAR [35]?

**Ethical Concerns:**

["NO or VERY MINOR ethics concerns only"]

**Final Justification:**

This work designs a more comprehensive prompt pool method for continual test-time adaptation and achieves better performance compared with the previous prompt pool-based method (DPCore [52]), albeit with increased computational burden. Given the comprehensive evaluation and the accuracy improvements demonstrated, I increase my rating to 4.

The reason I do not lean toward a higher score is that the “prompt pool” has been a classic technique in continual learning for several years, and this paper is not the first to apply it to continual test-time adaptation. While the technical improvement over DPCore is effective, it is not particularly impressive at the conceptual level.

**Limitations:**

yes

**Quality:**

3

**Strengths And Weaknesses:**

**Strengths**

1. The method overview figure is well-illustrated and effectively conveys the core idea.
2. The evaluations on benchmark datasets demonstrate that the proposed method outperforms recent baselines, indicating its effectiveness.


**Weaknesses**

1. The method appears technically incremental compared to previous visual prompt tuning-based CTTA approaches, such as DPCore [52]. It mainly extends DPCore by introducing a class prompt pool in addition to the domain prompt pool, and replaces the Online K-Means clustering in DPCore with MST-based clustering for prompt grouping. These seem like modest extensions, making it difficult to identify significant technical contributions.
2. Since the proposed method introduces additional operations during online adaptation and appears more complex than the baselines, it is important to clarify the performance-efficiency trade-off through a comprehensive computational analysis. However, the paper only reports learnable parameter counts, which is especially insufficient for prompt-tuning-based methods. I recommend including additional metrics such as memory usage, runtime, and computation cost (e.g., FLOPs).

---

> ### Author Rebuttal · Authors · 2025-07-30
>
> Thank you for your detailed review and recognition of our work’s strengths. Your valuable feedback on technical positioning relative to DPCore, comprehensive efficiency analysis, and clarifications on data stream construction (including model performance under different modes) is deeply appreciated. We’ve made additional experiments and analyses to address your concerns as well as strenghten the paper, detailed below:
>
> ### **1. Elaborate on the conceptual and technical contributions of the proposed method in comparison to DPCore.**
>
> We respectfully clarify that our contributions extend beyond incremental modifications to DPCore, with key distinctions addressing critical limitations in prior prompt-based CTTA methods:
>
> First, a correction: **DPCore does not employ Online K-Means clustering for prompts**, and its prompt pool grows unbounded without grouping mechanisms, leading to redundancy and noise accumulation. In contrast, our MST-based clustering is not a "replacement" but a novel design to actively merge similar prompts, which directly mitigates the parameter explosion and noise issues in DPCore.
>
> Second, the introduction of a class prompt pool is not merely an "additional" component. **It explicitly models class-level shared knowledge across domains, a dimension entirely absent in DPCore’s domain-only prompts.** This enables our method to leverage cross-domain class commonalities, a capability validated by our ablation study in Table 3, showing **significant performance gains (4.7%)** from this design.
>
> **Together with our knowledge fusion mechanism and class prompt introduction, these innovations form a cohesive framework that outperforms DPCore by 5.1%, which is a substantial improvement.** These designs address challenges in prompt-based CTTA (scalability and knowledge utilization), establishing clear technical contributions.
>
> ### **2. A more comprehensive efficiency analysis beyond the limited parameter count comparison.**
>
> We fully agree that a comprehensive computational analysis is critical to clarify the performance-efficiency trade-off, especially for prompt-tuning-based methods. To address this, we have supplemented detailed comparisons across multiple metrics, including learnable parameters, memory usage, runtime, and FLOPs, evaluated on a single NVIDIA 4090 GPU . Results are presented in two tables: one for ImageNet-to-ImageNet-C (first-round adaptation) and another for the 10th round under the repeating domains setting.
>
> Our findings demonstrate that despite introducing additional operations during online adaptation, **our method achieves a favorable balance between performance and efficiency:**
>
> - Compared to baselines like CoTTA and ViDA, our method leads in both efficiency and performance.
> - Against DPCore, we achieve a 5.1% performance gain with comparable efficiency metrics.
> - Compared to Tent, which only modifies batch normalization layers, has superior raw efficiency, our method achieves a 15.2% higher performance gain with little additional cost, a worthwhile trade-off.
>
> **A key efficiency advantage emerges in long-term adaptation:** In later rounds (e.g., R10), our KFI module stops generating new prompts when no new domains appear, reducing computational load over time. In contrast, DPCore—lacking fusion and prompt selection—suffers from escalating parameters, FLOPs, and runtime due to unconstrained prompt accumulation, alongside rising error rates. This dynamic efficiency gain further validates our design’s practicality for continuous adaptation scenarios.
>
> These results confirm that our method’s additional operations are justified by its performance gains and long-term efficiency stability, making it suitable for real-world deployment.
>
> Table A: Computational analysis on ImageNet-to-ImageNet-C.
> |Methods|Learnable Params|Relative Time|Memory Usage|Avg TFLOPs/batch|Err Mean|
> |-|-|-|-|-|-|
> |Tent|0.03M|1.0|5.5GB|1.08|51.0|
> |CoTTA|86.57M|4.7|16.2GB|3.24|49.9|
> |ViDA|93.70M|5.3|9.3GB|14.03|43.4|
> |DPCore|0.08M|1.6|5.7GB|1.67|39.9|
> |Ours|0.09M|1.9|6.0GB|1.68|34.8|
>
> Table B: Computational analysis on ImageNet-to-ImageNet-C with repeating domains, round 10.
> |Methods|Learnable Params|Relative Time|Memory Usage|Avg TFLOPs/batch|Err Mean|
> |-|-|-|-|-|-|
> |Tent|0.03M|1.0|5.5GB|1.08|99.9|
> |DPCore|1.03M|2.1|8.4GB|1.73|46.8|
> |Ours|0.20M|1.8|6.1GB|1.60|34.5|
>
> ### **3. Clarifications on data stream construction and evaluations considered temporally correlated data streams.**
>
> **Following common practices in CTTA, our data stream is constructed via independent and identically distributed (i.i.d.) sampling, consistent with regular training protocols.** This aligns with most baselines to ensure fair comparisons.
>
> Additionally, we further supplemented experiments on temporally correlated data streams, **following the "practical TTA" setting in RoTTA[47]**. Results in Table C show that our method achieves a further **4.7% improvement** in this scenario.
>
> This performance gain stems from the ability of our knowledge fission and fusion mechanisms to model temporal dependencies: fission captures time-varying knowledge, while fusion preserves stable temporal patterns across sequential data. These results validate the generalization of our method to real-world temporally correlated streams.
>
> Table C: Average error rate(%) over practical TTA settings on ImageNet-to-ImageNet-C.
> |Methods|Published|Err Mean|
> |-|-|-|
> |Source|-|55.8|
> |RoTTA|CVPR'23|49.5|
> |SAR|ICLR'23|44.0|
> |DPCore|ICML'25|43.9|
> |Ours|-|**39.2**|

---

> ### Author Response · Authors · 2025-08-05
> **Thanks and a kind reminder for discussion**
>
> Dear Reviewer 8gyU,
>
> Thank you for your time and effort in reviewing our work. As the rebuttal period concludes within three days, we wish to respectfully confirm whether our responses have adequatelly addressed your initial concerns.
>
> Any feedback would be greatly appreciated. We truly value your expertise and are happy to continue the discussion if you have further questions.
>
> Sincerely,
>
> The Authors

---

> ### Comment · Reviewer_8gyU · 2025-08-05
>
> Dear Authors,
>
> Thank you for your clarification and the additional numbers provided!
> I acknowledge that this work designs a more comprehensive prompt pool method and achieves better performance than the previous prompt pool-based method (DPCore [52]), albeit with increased computational burden. Given the comprehensive evaluation and the accuracy improvements demonstrated, I have decided to increase my score by 1.
>
> The reason I do not lean toward a higher score is that the “prompt pool” has been a classic technique in continual learning for several years, and this paper is not the first to apply it to continual test-time adaptation. The technical improvement over DPCore, while effective, does not significantly impress me at the conceptual level.
>
> Best regards,
>
> Reviewer 8gyU

---

> ### Author Response · Authors · 2025-08-06
> **Thank you for your feedback**
>
> Dear Reviewer 8gyU,
>
> Thank you sincerely for your thoughtful reconsideration and the increased score. We deeply appreciate your candid feedback on the conceptual positioning of our prompt pool design, which provides valuable perspective for our future research. We will continue to explore more conceptually impactful innovations in this direction, building on the foundations laid here.
>
> Thank you again for your time and insights.
>
> Sincerely,
>
> The Authors

---

### Official Review · Reviewer_TUaD · 2025-07-01

**Clarity:** 3
**Significance:** 3
**Originality:** 3
**Rating:** 4
**Confidence:** 4

**Summary:**

While prior CTTA methods mainly focus on mitigating catastrophic forgetting, they tend to suffer from insufficient learning of new domain knowledge and interference from outdated or harmful historical information. To address these limitations in CTTA, the authors propose KFF, a class-aware method that dynamically expands and merges domain knowledge at test time. The proposed method achieves strong performance with a low number of parameters, as demonstrated by extensive experiments on the ImageNet-C benchmark.

**Questions:**

- Misc) Could you explain more about Fig1 (c)? Which accuracy was decreased? I already got the authors point, but still have some vague point.
- For others, please refer to weakness.

**Ethical Concerns:**

["NO or VERY MINOR ethics concerns only"]

**Final Justification:**

Thanks for the author's rebuttal and effort. My concern regarding efficiency and repetitive scenarios has been resolved. I will maintain my current score, considering other reviewers' comments. I recommend that authors add more comparison with EATA and BECoTTA in their final version of the paper.

**Limitations:**

Yes

**Paper Formatting Concerns:**

They don't have formatting issues.

**Quality:**

3

**Strengths And Weaknesses:**

1. Strengths

- This paper is overall well-organized, well-motivated, and easy to follow. The authors highlight an important point: adaptation to new domains can sometimes be more crucial than preserving knowledge from past domains—an aspect that current CTTA works often overlook. Their proposed knowledge fusion/fission approach based on prompt learning is novel and intuitive.
- The experimental results are thorough and informative. Additionally, the proposed model is parameter-efficient compared to other CTTA baselines (similar to Tent), while achieving the lowest error rate. This is impressive.

2. Weaknesses

- While the model is indeed parameter-efficient, parameter number is not the only metric for efficiency. Could the authors also provide a comparison of inference time and memory usage with other baselines? For example, Tent is known for being fast and memory-efficient (from my experience), as it only modifies the batch normalization layers. In particular, comparisons with other prompt-based CTTA models would offer valuable insight.

- In Table 2, KFF performs well in maintaining initial performance (R1) compared to other baselines. However, the repeated performance levels appear to be merely preserved without noticeable improvement. Is this due to a strong reliance on source domain statistics? As I understand it, KFF focuses on acquiring new domain knowledge more actively than other methods, making the knowledge fission mechanism particularly relevant. A deeper analysis from the authors on this point would be appreciated.

---

> ### Author Rebuttal · Authors · 2025-07-30
>
> Thank you sincerely for your thoughtful assessment of our KFF method and your recognition of its strengths. Your constructive feedback, particularly regarding efficiency metrics beyond parameters and the analysis of repeated domain performance, is incredibly helpful. Addressing these,
> we’ve made additional experiments and clarifications to strengthen the paper, as detailed below:
>
> ### **1. More comparision on efficiency.**
>
> We have supplemented comparisons of learnable parameter count, GPU memory usage, and computational efficiency with baselines, evaluated on a single NVIDIA 4090 GPU. Results are provided in two tables: table A for the first round and table B for the 10th round under the repeating domains' setting.
>
> 1. **Comparison with Tent.**
> Tent, which only modifies batch normalization layers, exhibits superior raw efficiency (faster computation, lower memory). However, our method delivers a 15.2% performance improvement with little additional cost, making this trade-off worthwhile.
>
> 2. **Comparison with other baseline models.**
> Against CoTTA and ViDA, our method outperforms them in both efficiency and performance.
> Against DPCore, our method achieves a 5.1% performance gain while maintaining comparable efficiency: computation cost, memory usage, and parameter count remain on par in early rounds.
>
> 1. **Advantage of long-term adaptation.**
> A critical efficiency edge emerges in later rounds (R10, shown in Table B): our KFI module stops generating new prompts when no new domains appear, reducing computational load over time. In contrast, DPCore, lacking fusion and prompt selection, sees escalating parameters, slower inference, and higher memory usage due to unconstrained prompt accumulation (accompanied by rising error rates). This validates our design’s efficiency and stability in continual test-time adaptation.
>
> Overall, our method balances performance gains with efficiency, outperforming both prompt-based and non-prompt-based baselines in performance while maintaining comparable efficiency, and offering far greater performance improvements than ultra-lightweight methods like Tent with minimal additional cost.
>
> Table A: Computational analysis on ImageNet-to-ImageNet-C.
> |Methods|Learnable Params|Relative Time|Memory Usage|Avg TFLOPs/batch|Err Mean|
> |-|-|-|-|-|-|
> |Tent|0.03M|1.0|5.5GB|1.08|51.0|
> |CoTTA|86.57M|4.7|16.2GB|3.24|49.9|
> |ViDA|93.70M|5.3|9.3GB|14.03|43.4|
> |DPCore|0.08M|1.6|5.7GB|1.67|39.9|
> |Ours|0.09M|1.9|6.0GB|1.68|34.8|
>
> Table B: Computational analysis on ImageNet-to-ImageNet-C with repeating domains, round 10.
> |Methods|Learnable Params|Relative Time|Memory Usage|Avg TFLOPs/batch|Err Mean|
> |-|-|-|-|-|-|
> |Tent|0.03M|1.0|5.5GB|1.08|99.9|
> |DPCore|1.03M|2.1|8.4GB|1.73|46.8|
> |Ours|0.20M|1.8|6.1GB|1.60|34.5|
>
> ### **2. A deeper analysis on why the repeated performance levels appear to be merely preserved without noticeable improvement.**
>
> **The stable performance across repeated rounds does not stem from the strong reliance on source statistics.** **In our setup, R1 samples already provide sufficient information for the model to capture robust patterns, establishing a strong baseline, and our methods could maintain that performance.** This is a deliberate strength, as many baselines struggle to even maintain initial performance due to the inevitable mixing of conflict domain knowledge (e.g., Tent drops by 48.9% from R1 to R10, while DPCore drops by 6.9% in Table 2 in our paper).
>
> Our knowledge fission/fusion mechanisms focus on retaining and refining knowledge rather than forcing unnecessary "improvement" when domains stabilize. When we reduced per-domain samples to 1/10, as shown in Table C, clear improvements emerged across rounds, showcasing KFF’s ability to actively acquire new knowledge when needed. This validates its balance of stability (with sufficient data) and adaptability (with limited data).
>
> Table C: Classification error rate(%) for ImageNet-to-ImageNet-C online CTTA task in 10 repeated rounds with limited samples(R1-R10).
> |Methods|R1|R2|R3|R4|R5|R6|R7|R8|R9|R10|Avg|
> |-|-|-|-|-|-|-|-|-|-|-|-|
> |Source|50.5|50.5|50.5|50.5|50.5|50.5|50.5|50.5|50.5|50.5(+0.0%)|50.5|
> |CoTTA|49.8|49.9|50.1|50.2|50.3|50.6|50.6|50.8|50.9|50.9(-1.1%)|50.4|
> |ViDA|46.6|43.7|42.5|42.3|42.8|42.6|42.4|42.3|42.2|42.1(+4.5%)|43.0|
> |DPCore|49.9|46.6|45.7|45.8|45.6|45.3|45.7|45.4|45.4|45.3(+4.5%)|46.1|
> |Ours|43.4|40.2|39.1|38.3|37.6|37.1|36.7|36.5|36.5|**35.8(+6.9%)**|**38.1**|
>
> ### **3. Explain more about Fig1 (c).**
>
> Fig. 1(c) illustrates the performance of existing methods during a domain change: fog → bright → fog. Specifically, **when the model encounters the fog domain again (after adapting to the bright domain), its classification accuracy drops sharply by 14.6% compared to the first fog domain**, even when tested on identical fog samples.
>
> This decline indicates that knowledge learned from the bright domain conflicts with prior knowledge from the fog domain, impairing the model’s ability to reuse previously acquired fog-specific patterns. In contrast, our method mitigates such conflicts through knowledge fission strategy, preserving accuracy across repeated domains, which further validates its effectiveness.

---

> > ### Comment · Reviewer_TUaD · 2025-08-05
> >
> > Thanks for the author's rebuttal and effort. I will maintain my score considering other reviewers' comments.

---

> ### Author Response · Authors · 2025-08-05
> **Thanks for your invaluable feedback!**
>
> Dear Reviewer TUaD,
>
> Thanks again for your expertise and invaluable feedback on improving the quality of our paper!
>
> Best Regards,
>
> The Authors

---

### Official Review · Reviewer_m3Qy · 2025-07-02

**Clarity:** 3
**Significance:** 3
**Originality:** 3
**Rating:** 4
**Confidence:** 4

**Summary:**

The paper presents a Class-aware Domain Knowledge Fusion and Fission (KFF) framework for Continual Test-Time Adaptation (CTTA), addressing critical challenges of catastrophic forgetting and inefficient new knowledge assimilation. The key idea is to adaptively expand and merge class-aware domain knowledge from old and new domains based on test-time data, allowing discriminative historical knowledge to accumulate dynamically. The authors evaluate KFF on datasets like ImageNet-C, demonstrating its effectiveness against other methods in both non-repeating and repeating domain scenarios.

**Questions:**

1.	In the class prompt pool, not all potential labels are saved, as similar classes are clustered and updated with weighted averaging. Does this have an impact? In other words, if the prompt pool were structured to save prompts for potentially all categories based on pseudo-labels (with the pool size fixed to the total number of labels), would the performance decrease?

2.	All experiments are based on Vision Transformer. Can the KFF method be transferred to other models such as ResNet or MobileNet?

**Ethical Concerns:**

["NO or VERY MINOR ethics concerns only"]

**Final Justification:**

The authors provide more clarification to demonstrate the robustness of hyper-parameters, and commit to including more discussions with related works. I maintain positive toward the paper.

**Limitations:**

The authors adequately addressed the limitations in the paper.

**Quality:**

3

**Strengths And Weaknesses:**

**Strengths**:

1.	The paper addresses challenges of catastrophic forgetting and insufficient new knowledge assimilation in CTTA.
2.	It proposes a novel class-aware knowledge fusion and fission framework for adaptive knowledge management.
3.	The method demonstrates state-of-the-art performance with high parameter efficiency in various continual test-time adaptation scenarios.

**Weaknesses**:

1.	More comparative experiments are needed. Methods applicable to CTTA scenarios, such as EATA[1] and BECoTTA[2], should be included in the comparative experiments.
2.	A comparison with knowledge fusion frameworks similar to KFF is necessary. A recent work, CoLA[3], proposes a plug-and-play TTA framework that performs knowledge fusion at the weight level. It could potentially achieve better improvements in CTTA scenarios when combined with existing baseline methods.
3.	Lack of deeper analysis on knowledge fusion. While fusion can prevent the prompt pool from expanding indefinitely, it might also lead to the loss of subtle knowledge potentially valuable for future adaptation. The current ablation studies on N_d and N_c are too limited in scope, requiring more extensive analysis.
4.	Ablation experiments need to be supplemented. KFF only updates prompts whose entropy values are below a certain threshold. Please provide an ablation study for this threshold.
5.	The “Domain Knowledge Fission” section appears to share similarities with DPCore[A]. Please explain in more detail the differences between this part and DPCore.
6.	The theoretical analysis (Section A) assumes well-separated clusters with intra-cluster distances smaller than inter-cluster distances. However, domain distributions in real-world scenarios often overlap. It would be helpful to analyze KFF’s performance under such overlapping scenarios to better validate its practicality.
7.	Experiments are primarily conducted on synthetic corrupted datasets (e.g., ImageNet-C, CIFAR100-C), with limited validation on real-world complex distribution shifts (e.g., varying illumination, weather, or device-captured natural images). Supplementing such experiments would strengthen the demonstration of practical utility.

[A] DPCore: Dynamic Prompt Coreset for Continual Test-Time Adaptation. ICML 2025.

[1] Efficient Test-Time Model Adaptation without Forgetting. ICML 2022.

[2] BECoTTA: Input-dependent Online Blending of Experts for Continual Test-time Adaptation. ICML 2024.

[3] Cross-Device Collaborative Test-Time Adaptation. NIPS 2024.

---

> ### Author Rebuttal · Authors · 2025-07-30
>
> Thank you for your detailed evaluation of our KFF framework and recognition of its strengths. The constructive points you raised are highly valuable. To address these, we conducted supplementary experiments and analyses detailed below:
>
> ### **1. More comparative experiments.**
>
> We further supplemented experiments comparing our method with EATA[34] and BECoTTA[24] on the ImageNet-to-ImageNet-C benchmark, with results presented in Table A.
>
> It can be seen that our method achieves an average error rate reduction of **more than 10%** compared to both EATA and BECoTTA. This performance gain can be attributed to the limitations of the competing methods: EATA and BECoTTA do not account for the dynamic evolution of domain knowledge, which reduces the efficiency of knowledge utilization. In contrast, our knowledge fission and fusion mechanism not only significantly controls model size but also promotes the integration of similar knowledge, thereby enhancing the generality of learned representations. Additionally, our class-specific prompts and selective fusion strategy better leverage shared features of the same/similar classes across domains, further boosting classification accuracy. These results provide additional evidence for the effectiveness of our approach in CTTA scenarios.
>
> Table A: Average error rate(%) for ImageNet-to-ImageNet-C CTTA task across different methods.
> |Methods|Publish|Err Mean|
> |-|-|-|
> |Source|-|55.8|
> |EATA|ICML'22|45.4|
> |BECoTTA|ICML'24|47.7|
> |Ours|-|**34.8**|
>
> ### **2. Comparison with knowledge fusion frameworks similar to KFF.**
>
> We have conducted two supplementary experiments by integrating the knowledge fusion module proposed in CoLA[a]: (1) integrating CoLA’s knowledge fusion module into our baseline DPCore; (2) replacing our KFU module with CoLA’s fusion mechanism. The results were presented in Table B, which show that incorporating CoLA into the baseline yields only a 0.8% improvement, while replacing our KFU with CoLA leads to a 3.0% gain—both significantly lower than our method’s 5.1% enhancement. **Compared to these two setups, our approach achieves additional gains of 4.3% and 2.1%, respectively.**
>
> The gap arises because CoLA aggregates weights unoptimizedly based solely on domain similarity, ignoring class-level relationships. In contrast, our KFF uses global similarity to merge both domain- and class-specific knowledge, better leveraging cross-domain shared features of same/similar classes—validating its superiority.
>
> Table B: Average error rate(%) for ImageNet-to-ImageNet-C CTTA task across different fusion strategies.
> |Methods|Err Mean|
> |-|-|
> |Source|55.8|
> |DPCore|39.9|
> |DPCore+CoLA|39.1|
> |Ours|**34.8**|
> |Ours+CoLA|36.9|
>
> [a] Cross-Device Collaborative Test-Time Adaptation. NeurIPS 2024.
>
> ### **3. Deeper analysis on knowledge fusion.**
>
> We have extended analyses on how N_d and N_c balance pool expansion control and retention of subtle, future-valuable knowledge. Experiments, spanning a broader parameter range under the "CTTA with Repeating Domains" setting (R1 and R10 for the first and 10th round), are in Tables C and D.
>
> Results show that **excessively small N restrict prompt diversity, losing subtle knowledge critical for future adaptation, while overly large N cause uncontrolled pool expansion, leading to redundant prompts poorly optimized by limited samples.**
>
> Our method selects optimal N_d/N_c, balancing these trade-offs to avoid both indefinite expansion and valuable knowledge loss, performing best in both R1 and R10.
>
> Notably, in R1, large N_d/N_c yield stable performance because the Knowledge Fission Module (KFI) generates too few prompts to trigger fusion, letting the framework dynamically avoid unnecessary early fusion and preserve initial subtle knowledge.
>
> In summary, our extended analyses confirm that properly tuned N_d and N_c effectively prevent both unchecked pool expansion and loss of valuable subtle knowledge, validating the robustness of our knowledge fusion design.
>
> Table C: Effect of N_c. Average error rate(%) for ImageNet-to-ImageNet-C CTTA task.
> |N_c|50|100*|150|200|500|1000|2000|
> |-|-|-|-|-|-|-|-|
> |R1|34.9|**34.8**|**34.8**|**34.8**|35.2|35.2|35.2|
> |R10|35.1|**34.5**|34.7|34.8|35.3|35.5|36.6|
>
> Table D: Effect of N_d. Average error rate(%) for ImageNet-to-ImageNet-C CTTA task.
> |N_d|10|20*|40|50|60|100|200|
> |-|-|-|-|-|-|-|-|
> |R1|35.1|**34.8**|34.9|36.1|36.1|36.1|36.1|
> |R10|35.1|**34.5**|34.8|36.3|36.8|36.8|37.0|
>
> ### **4. Ablation experiments for γ_h.**
>
> We further conducted an ablation study on the entropy threshold (γ_h), with results in Table E. The analysis reveals that γ_h impacts performance through two key mechanisms: **While too low γ_h reduces valid samples for updates, hindering adaptation, too high γ_h includes high-entropy (low-confidence) samples as noise, hurting performance**. This finding is consistent with EATA [34], which also demonstrated that adaptation on test samples with very high entropy may hurt performance.
>
> Our method selects an optimal γ_h that balances these trade-offs, ensuring sufficient valid samples for updates while avoiding noise from high-entropy instances, thereby achieving robust performance. This ablation study confirms the effectiveness of our threshold selection strategy.
>
> Table E: Effect of γ_h, we choose 2 in our paper. Average error rate(%) for ImageNet-to-ImageNet-C CTTA task.
> |γ_h|1|1.5|2*|2.5|3|3.5|4|
> |-|-|-|-|-|-|-|-|
> |Ours|35.0|**34.8**|**34.8**|35.0|35.1|35.4|35.8|
>
> ### **5. Explain the differences between Domain Knowledge Fission and DPCore.**
>
> In terms of Domain Knowledge Fission, **the core distinction from DPCore lies in our Domain Prompt Selection Strategy**:
>
> DPCore aggregates all existing domain prompts via similarity-weighted fusion during adaptation, which inevitably introduces noise from irrelevant domain prompts. In contrast, our Domain Knowledge Fission selectively picks only the most relevant domain prompts to guide the generation of new domain prompts, significantly reducing noise interference. A controlled experiment (Table F) shows that adopting DPCore’s full domain prompt weighting strategy ("Ours*") leads to a 1.8% performance drop, validating the effectiveness of our selective domain prompt selection in fission.
>
> Beyond fission, our method differs from DPCore in two other key aspects: (1) We introduce class prompts (absent in DPCore) to explicitly capture cross-domain class-level commonalities, boosting performance by 4.7% (ablation in Table 3). (2) Our prompt fusion merges similar prompts to avoid redundancy, whereas DPCore’s prompt pool grows unbounded, alleviating noise accumulation and parameter explosion.
>
> Table F: Performance Impact of different Domain Prompt Selection Strategies: Ours vs. DPCore-style (denoted as Ours*).
> |Methods|Err Mean|
> |-|-|
> |Source|55.8|
> |DPCore|39.9|
> |Ours|**34.8**|
> |Ours*|36.6|
>
> ### **6. Analyze KFF’s performance under overlapping scenarios.**
>
> We supplemented experiments on overlapping domain distributions, following CCC[38] to construct scenarios with controlled feature overlap between domains.
>
> Results (Table G) show our method outperforms previous SOTA DPCore by **3.7%** on average under such conditions. This shows our knowledge fission/fusion mechanisms can still accurately distinguish and integrate knowledge types despite overlaps, maintaining strong generalization, validating KFF’s practicality for real-world scenarios.
>
> Table G: Average error rate(%) on ImageNet-C, CCC-Medium and CCC-Hard.
> |Methods|ImageNet-C|CCC-Medium|CCC-Hard|
> |-|-|-|-|
> |Source|55.8|58.0|78.0|
> |RDumb|45.8|49.8|76.2|
> |CoTTA|49.9|52.5|79.5|
> |DPCore|39.9|43.2|74.1|
> |Ours|**34.8**|**37.2**|**72.7**|
>
> ### **7. Experiments on real-world complex distribution shifts.**
>
> We validated the effectiveness of our method on datasets with natural domain shifts (Cityscapes-to-ACDC). The experimental results are shown in Table H, which demonstrate that **our method outperforms the existing SOTA method (DPCore) by 1.1%**. This is because our knowledge fusion and fission can adaptively adjust according to the shift degree of different domains and select the most appropriate prompt for optimization, which indicates that our knowledge fission and knowledge fusion still maintain high generalization on datasets with real-world complex distribution shifts.
>
> Table H: Average mIoU score(%) for Cityscapes-to-ACDC, with SegFormer-B5 model pre-trained on the Cityscapes dataset as backbone.
> |Methods|round1|round2|round3|Mean|
> |-|-|-|-|-|
> |Source|56.7|56.7|56.7|56.7|
> |Tent|56.7|55.9|55.0|55.7|
> |CoTTA|58.6|58.6|58.6|58.6|
> |ViDA|61.1|62.2|62.3|61.9|
> |DPCore|62.1|62.6|62.8|62.5|
> |Ours|**63.1**|**63.6**|**64.0**|**63.6**|
>
> ### **Q1. Would the performance decrease if the prompt pool were based on pseudo-labels?**
>
> **Yes, saving all class prompts via pseudo-labels reduces performance: Appendix Table 4 shows a 1% accuracy drop**, as pseudo-labels fail to capture shared traits among similar classes, resulting in suboptimal knowledge representation. By contrast, our design leverages inter-class commonalities, achieving the highest accuracy among compared approaches.
>
> ### **Q2. Can the KFF method be transferred to other models such as ResNet or MobileNet?**
>
> Our KFF method, like many other prompt-based approaches (e.g. DePT[12], DPCore[52]), is initially designed around Vision Transformers (ViTs)[10, b] due to their token-based structure, which naturally supports prompt integration. **Conventional CNNs (e.g., ResNet, MobileNet) lack such token mechanisms, making direct adaptation of existing prompt frameworks challenging.**
>
> To align with common practices in CTTA and ensure fair comparisons (most SOTAs[25, 26, 52] use ViTs), we adopt ViTs as the backbone. However, we plan to explore CNN-compatible prompt designs in future work to verify KFF’s transferability across architectures.
>
> [b] SegFormer: Simple and Efficient Design for Semantic  Segmentation with Transformers. NeurIPS 2021.

---

> > ### Comment · Reviewer_m3Qy · 2025-08-04
> >
> > Thank you for the reply.
> >
> > For follow-up discussion, it seems that you try to integrate CoLA into DPCore/KFF via its similarity-based knowledge aggregation scheme in the prior rebuttal. I am curious if the backpropagation-based knowledge reprogramming scheme in CoLA can also be applied in KFF to learn the aggregation weight of different prompts? This may be more efficient, by reducing an unnecessary forward pass used solely for similarity measurements, and can be more effective, by learning the optimal aggregation to directly minimize the objectives. Could the authors provide more discussion or results on this perspective? Besides, CoLA is a strong baseline for the online CTTA task with repeated rounds, and it is thus highly recommended to include CoLA in Table 2 of the main paper for a more rigorous comparison.
> >
> > Moreover, using both feature alignment and entropy minimization as the test-time objectives is exactly the same as FOA [1], and it is suggested to include a citation/discussion to FOA in Section 3.2.
> >
> > [1] Test-Time Model Adaptation with Only Forward Passes. ICML 2024.

---

> > > ### Author Response · Authors · 2025-08-06
> > > **Thanks for your feedback and further response**
> > >
> > > Dear Reviewer m3Qy,
> > >
> > > We deeply appreciate your expertise and detailed feedback. We provide further discussion and results  below.
> > >
> > > 1. We have experimented with applying CoLA's backpropagation-based knowledge reprogramming to KFF, with results in Table Ⅰ. In the experiments, we tried three different applications: Ours$^1$ (integrating CoLA's similarity-based aggregation), Ours$^2$ (adopting CoLA's backpropagation scheme and reduce one forward pass), and Ours$^3$ (applying backpropagation scheme but do not reduce forward pass for the class prompt). **While CoLA's scheme reduces one forward pass, it has limitations: it fails to optimize for intra-domain variations and cannot extend to class prompts** (critical for capturing cross-domain class-shared features). In contrast, our method achieves more accurate knowledge fusion, outperforming Ours$^2$ by 5% and delivering 2% gains over Ours$^1$ and Ours$^3$ with the same computational cost.
> > >
> > > 2. We have supplemented results of KFF vs. CoLA under repeating domains in Table Ⅱ. Although CoLA can retain historical knowledges, it still lags behind our method in repeated rounds, as it struggles with two key issues: **(i) the backpropagation-based aggregation weights are insensitive to intra-domain variations, leading to suboptimal utilization of historical knowledge; (ii) without explicit class-level modeling, it fails to leverage shared features across similar classes in repeated domains.** In contrast, our fusion-fission mechanism addresses both by dynamically selecting prompts and preserving class-specific knowledge, achieving more than 2.5% performance gain with comparable efficiency, enabling better adaptation across repeated rounds.
> > >
> > > 3. We will add a citation to FOA in Section 3.2, clarifying key differences:
> > >   - FOA optimizes input prompts via derivative-free evolution strategies, while KFF tunes learnable prompt tokens with gradient descent (more sample-efficient for CTTA).
> > >   - Our fusion-fission mechanism dynamically balances new knowledge acquisition and historical knowledge retention, which is absent in FOA.
> > >
> > > Table Ⅰ: Average error rate(%) for ImageNet-to-ImageNet-C CTTA task.
> > > |Methods|Class Prompt|Err Mean|
> > > |-|-|-|
> > > |Source|✔|55.8|
> > > |DPCore|-|39.9|
> > > |Ours|✔|**34.8**|
> > > |Ours$^1$|✔|36.9|
> > > |Ours$^2$|-|39.8|
> > > |Ours$^3$|✔|37.1|
> > >
> > > Table Ⅱ: Classification error rate(%) for ImageNet-to-ImageNet-C  CTTA task in 10 repeated rounds.
> > > |Methods|R1|R2|R3|R4|R5|R6|R7|R8|R9|R10|Avg|
> > > |-|-|-|-|-|-|-|-|-|-|-|-|
> > > |Source|55.8|55.8|55.8|55.8|55.8|55.8|55.8|55.8|55.8|55.8|55.8|
> > > |DPCore+CoLA|39.1|39.0|38.5|38.6|38.4|38.4|38.3|38.3|38.3|38.3|38.5|
> > > |Ours|34.8|34.6|34.6|34.6|34.3|34.2|34.4|34.4|34.4|34.5|**34.5**|
> > > |Ours$^1$|36.9|36.8|36.8|36.7|36.9|37.1|37.2|37.1|37.0|37.0|37.0|
> > > |Ours$^2$|39.8|39.5|39.4|39.4|39.3|39.1|39.2|39.2|39.2|39.2|39.3|
> > > |Ours$^3$|37.1|36.9|36.8|36.7|36.8|37.0|37.1|37.2|37.2|37.1|37.0|
> > >
> > > Sincerely,
> > >
> > > The Authors

---

> > > > ### Comment · Reviewer_m3Qy · 2025-08-06
> > > >
> > > > Thank you for your additional clarifications. I will maintain my positive score.
> > > >
> > > > Please ensure to include the results of CoLA (with EATA/DeYO) for comparisons on the CTTA task with 10 repeated rounds in the revised paper.

---

> > > > > ### Author Response · Authors · 2025-08-06
> > > > > **Thanks for your invaluable feedback**
> > > > >
> > > > > Dear Reviewer m3Qy,
> > > > >
> > > > > Thank you for your positive feedback and guidance. We will include the results of CoLA (with EATA/DeYO) for comparisons in the revised paper. Thank you again for your expertise and invaluable feedback on improving the quality of our paper!
> > > > >
> > > > > Sincerely,
> > > > >
> > > > > The Authors

---

### Official Review · Reviewer_er6C · 2025-07-03

**Clarity:** 3
**Significance:** 2
**Originality:** 2
**Rating:** 5
**Confidence:** 4

**Summary:**

The paper proposes a method for continual test-time adaptation (CTTA) that fine-tunes model behavior during the test phase while accounting for potential domain shifts in the input stream. The core idea involves dynamically managing domain knowledge through a Knowledge Fission (KFI) process, which enables the model to adapt to new domains without forgetting previously seen ones. To prevent unbounded growth of the domain knowledge pool, the authors introduce a complementary Domain Fusion (DFU) mechanism that merges similar domain knowledge representations to maintain a manageable size. Adaptation is achieved using learnable domain and class prompt tokens, which store new knowledge while keeping the backbone of the pretrained ViT classifier frozen throughout the test-time adaptation process.

**Questions:**

I have two main concerns, both of which, if properly addressed, would significantly strengthen the paper and could change my evaluation from a borderline accept to an accept:

Evaluation on Larger, Real-World CTTA Benchmarks:
The current evaluation is primarily based on ImageNet-C and two small-scale toy datasets (CIFAR10/100-C). To better demonstrate the method’s scalability and real-world applicability, I strongly recommend including experiments on large-scale CTTA benchmarks with natural domain shifts, such as Cityscapes-to-ACDC. This would also allow for a more meaningful comparison with DPCore, which reports results on this benchmark.

Sensitivity and Robustness of Hyperparameters (γ and α):
The method introduces several critical hyperparameters (γ_d, γ_c, γ_h, α_d, α_c) that govern domain knowledge fission and fusion. While some sensitivity analysis is presented for prompt pool size, it would be important to assess how robust these γ and α parameters are across different datasets. A sensitivity study or discussion on their generalizability would greatly improve confidence in the method’s stability and ease of deployment.

**Ethical Concerns:**

["NO or VERY MINOR ethics concerns only"]

**Final Justification:**

The author response addressed most of my concerns in my initial review by providing additional experiments on the Cityscapes-to-ACDC dataset as well as more extensive hyperparameter analysis.  Thus, I change my final rating from a weak accept to an accept.

**Limitations:**

yes

**Quality:**

3

**Strengths And Weaknesses:**

Strengths
- The paper is clearly written, with well-articulated motivation and methodology, making it easy to follow.

- The proposed Knowledge Fission (KFI) and Domain Fusion (DFU) mechanisms significantly improve adaptation performance on ImageNet-to-ImageNet-C. This improvement is particularly notable in the repeating domains setting, where CTTA methods must preserve knowledge of previously seen domains.

- The method achieves state-of-the-art performance while maintaining a compact set of learnable parameters, avoiding unnecessary model bloat.

The paper includes ablation studies demonstrating the effectiveness of KFI and DFU, along with sensitivity analysis for the prompt pool size, providing valuable insights into the model's behavior.

Weaknesses
- The method is primarily evaluated on ImageNet-to-ImageNet-C, supplemented by CIFAR10 and CIFAR100, which are relatively small-scale and arguably toy datasets. More compelling evaluation on large-scale CTTA benchmarks with natural domain shifts—such as Cityscapes-to-ACDC, as reported in DPCore (a key competing method)—would strengthen the empirical evidence.

- The method introduces a large number of hyperparameters (e.g., γ_d, γ_c, γ_h, α_d, α_c, τ_d, τ_c, a, N_d, N_c). While sensitivity analysis was conducted for N_d and N_c, the influence of the remaining hyperparameters—especially γ and α parameters—on fission/fusion and knowledge retention is not discussed. This is particularly relevant because the method shows large gains on ImageNet-to-ImageNet-C with parameters tuned on a disjoint validation set from ImageNet-C. However, when transferred to CIFAR10 and CIFAR100 without further tuning, the gains over DPCore are minimal or marginal.

- While the KFI and DFU components are novel, the overall CTTA approach is clearly inspired by DPCore, particularly in the use of dynamic prompt-based adaptation. A more thorough comparison or differentiation from DPCore would be helpful to better highlight the novelty.

---

> ### Author Rebuttal · Authors · 2025-07-30
>
> Thank you sincerely for your insightful feedback and constructive suggestions, which have been invaluable in refining our work. We greatly appreciate your recognition of the method’s strengths in CTTA performance, parameter efficiency, and the clarity of our presentation. Addressing your key concerns, we have supplemented critical experiments and analyses to strengthen the paper, as detailed below:
>
> ### **1. Evaluation on Larger, Real-World CTTA Benchmarks.**
>
> Following the common practice in the CTTA community, we have conducted fair comparisons with existing methods on ImageNet-to-ImageNet-C, CIFAR10, and CIFAR100. In addition, we further validated the effectiveness of our method on datasets with natural domain shifts (Cityscapes-to-ACDC). The experimental results are shown in Table A. The results demonstrate that **our method outperforms the existing SOTA method (DPCore) by 1.1%**. This is because our knowledge fusion and fission can adaptively adjust according to the shift degree of different domains and select the most appropriate prompt for optimization. The result on Cityscapes-to-ACDC dataset indicates that our knowledge fission and knowledge fusion still maintain high generalization on datasets with natural domain shifts.
>
> Table A: Average mIoU score(%) for Cityscapes-to-ACDC, with SegFormer-B5 model pre-trained on the Cityscapes dataset as backbone.
> |Methods|Round1|Round2|Round3|Mean|
> |-|-|-|-|-|
> |Source|56.7|56.7|56.7|56.7|
> |Tent|56.7|55.9|55.0|55.7|
> |CoTTA|58.6|58.6|58.6|58.6|
> |ViDA|61.1|62.2|62.3|61.9|
> |DPCore|62.1|62.6|62.8|62.5|
> |Ours|**63.1**|**63.6**|**64.0**|**63.6**|
>
> ### **2. Analyze Sensitivity and Robustness of Hyperparameters (γ and α).**
>
> We conducted further analyses on the influences of key hyperparameters (γ and α) on knowledge fission/fusion, with results summarized in Table B-E. Numbers with * represent the original default parameter tuned on the disjoint validation set from ImageNet-C.
>
> For γ: As shown in cross-dataset experiments  (Table B,C), while the optimal γ value varies slightly between ImageNet-C, CIFAR10, and CIFAR100, **the performance remains stable within a reasonable range (±0.5%) across all datasets.** Excessively small or large γ degrades performance consistently: **a small γ introduces irrelevant knowledge, while a large γ limits knowledge reuse.** This consistent cross-dataset behavior highlights γ’s robustness: **though not universally optimal, its core tuning logic, balancing knowledge relevance and reuse, generalizes well, simplifying deployment via systematic adjustment rather than arbitrary search.**
>
> For α: **Across all tested datasets, α shows minimal impact on performance.** Our experiments (Table D,E) demonstrate that varying α within a typical range (e.g., 0–0.5) leads to performance fluctuations of **less than 0.2%** on ImageNet-C, CIFAR10, and CIFAR100. This insensitivity stems from the fact that our knowledge fission mechanism selects similar prompts for updates, promoting the reuse and evolution of similar knowledge while balancing the consistency of knowledge before and after fission. Thus, **hyperparameter α has minimal impact on consistent knowledge, further verifying the robustness and generalization of our method.**
>
> Regarding the performance on CIFAR10 and CIFAR100: We have organized additional experiments to clarify this, as shown in Table F (and Appendix Tables 5/6). Without further parameter tuning, our method already achieves improvements of 2.6% and 3% over DPCore on CIFAR10 and CIFAR100, respectively. These are not marginal gains.
>
> Furthermore, with simple parameter tuning (specifically adjusting γ_c to a larger value and N_c to a lower value) on CIFAR10 and CIFAR100, the average performance on the two datasets is further improved, resulting in an average gain of 3.3% over DPCore (denoted as "Ours+"). This adjustment is reasonable because CIFAR10/100 have smaller data scales compared to ImageNet-C; a stricter γ_c, through more rigorous knowledge matching, better models the discriminative knowledge between samples of different classes. This further demonstrates the generalization of our method across datasets of varying sizes.
>
> Table B: Effect of γ_d across different datasets. Average error rate(%) for CTTA tasks.
> ||10|15|20|25*|30|35|40|
> |-|-|-|-|-|-|-|-|
> |ImageNet-C|36.2|35.0|**34.8**|**34.8**|35.0|35.7|35.1|
> |CIFAR10-C|12.7|12.5|**12.4**|**12.4**|**12.4**|12.5|12.5|
> |CIFAR100-C|22.9|22.8|22.6|**22.5**|**22.5**|22.7|22.8|
>
> Table C: Effect of γ_c across different datasets. Average error rate(%) for CTTA tasks.
> ||1e-4|1e-3|5e-3*|7e-3|1e-2|5e-2|1e-1|
> |-|-|-|-|-|-|-|-|
> |ImageNet-C|36.0|35.2|**34.8**|**34.8**|35.1|35.2|35.8|
> |CIFAR10-C|12.7|12.6|12.4|12.4|12.3|**12.2**|**12.2**|
> |CIFAR100-C|23.0|22.7|22.5|22.5|22.4|**22.1**|22.3|
>
> Table D: Effect of α_d across different datasets. Average error rate(%) for CTTA tasks.
> ||0|0.05|0.1*|0.15|0.2|0.3|0.5|
> |-|-|-|-|-|-|-|-|
> |ImageNet-C|34.9|**34.8**|**34.8**|**34.8**|34.9|34.9|34.9|
> |CIFAR10-C|12.5|**12.4**|**12.4**|12.5|12.5|12.6|12.6|
> |CIFAR100-C|22.7|22.6|**22.5**|**22.5**|**22.5**|22.6|22.6|
>
> Table E: Effect of α_c across different datasets. Average error rate(%) for CTTA tasks.
> ||0|0.05|0.1*|0.15|0.2|0.3|0.5|
> |-|-|-|-|-|-|-|-|
> |ImageNet-C|34.9|**34.8**|**34.8**|**34.8**|**34.8**|34.9|35.0|
> |CIFAR10-C|12.6|12.5|**12.4**|**12.4**|12.5|12.6|12.6|
> |CIFAR100-C|22.7|22.7|**22.5**|**22.5**|22.6|**22.5**|22.6|
>
> Table F: Average error rate(%) for CTTA tasks across different datasets.
> |Methods|CIFAR10-C|CIFAR100-C|Average|
> |-|-|-|-|
> |Source|28.2|35.4|31.8|
> |DPCore|15.4|25.1|20.3|
> |Ours|12.4|22.5|17.5|
> |Ours+|**11.9**|**21.8**|**16.9**|
>
>
> ### **3. A more thorough comparison or differentiation from DPCore.**
>
> Our method differs from DPCore in three key aspects, with experimental evidence supporting the effectiveness of each innovation:
>
> 1. **Knowledge Fusion Mechanism:** Unlike DPCore, which allows unrestricted prompt growth (leading to escalating model size and potential redundancy), we introduce a knowledge fusion mechanism that merges similar prompts during adaptation. This not only controls parameter explosion but also promotes the integration of analogous knowledge, enhancing the generality of learned representations. As shown in our ablation study (Table 3 in our paper), this mechanism alone yields a **performance improvement of 2.1%**, validating its role in balancing efficiency and knowledge utility.
>
> 2. **Class-Specific Prompts:** DPCore relies on a single type of prompts. In contrast, we introduce class prompts to explicitly capture discriminative features shared by the same/similar classes across domains. This design enables the model to leverage cross-domain commonalities of classes for better classification. Our ablation results (Table 3) demonstrate that adding class prompts contributes **an additional 4.7% improvement**, highlighting its effectiveness in exploiting inter-domain class-level consistency.
>
> 3. **Dynamic Prompt Selection Strategy:** DPCore aggregates all existing prompts via similarity-weighted fusion for each test batch, which inevitably introduces noise from irrelevant prompts. Instead, our method selects only a small subset of the most relevant prompts for fusion, significantly reducing noise interference. To verify this, we conducted a controlled experiment where we modified our method to adopt DPCore’s full prompt weighting strategy (denoted as "Ours*"), shown in Table G. The observed performance dropped **1.8%** in "Ours*" compared to our original design, which confirms that our selective fusion strategy is critical for avoiding noise and maintaining robustness.
>
> These innovations collectively distinguish our approach from DPCore: by **controlling parameter growth, enhancing class-level knowledge utilization, and reducing noise when using prompts**, our method achieves superior performance as validated by both ablation studies and comparative experiments.
>
> Table G: Performance Impact of different Domain Prompt Selection Strategies: Ours vs. DPCore-style (denoted as Ours*)
> |Methods|Err Mean|
> |-|-|
> |Source|55.8|
> |DPCore|39.9|
> |Ours|**34.8**|
> |Ours*|36.6|

---

> ### Author Response · Authors · 2025-08-05
> **Thanks and a kind reminder for discussion**
>
> Dear Reviewer er6C,
>
> Thank you for your time and effort in reviewing our work. As the rebuttal period concludes within three days, we wish to respectfully confirm whether our responses have adequatelly addressed your initial concerns.
>
> Any feedback would be greatly appreciated. We truly value your expertise and are happy to continue the discussion if you have further questions.
>
> Sincerely,
>
> The Authors

---

> ### Author Response · Authors · 2025-08-08
> **Thank you and looking forward to your feedback**
>
> Dear Reviewer er6C,
>
> Thank you again for your invaluable feedback on improving the quality of our paper. We hope our responses have addressed your initial concerns thoroughly. Could you please kindly share your feedback on the supplementary experiments and clarifications provided? Thank you for your time and consideration.
>
> Sincerely,
>
> The Authors

---

### Note · Authors · 2025-08-13

Dear Reviewers, Area Chairs, and Program Chairs,

We would like to express our most sincere gratitude to all reviewers, area chairs, and program chairs. Your valuable suggestions have played a crucial role in further improving our paper.

First, we greatly appreciate all the Reviewers for their recognition of the clarity of our presentation, the effectiveness of our proposed Knowledge Fission and Fusion mechanisms, and the strong experimental performance on CTTA tasks. Meanwhile, we sincerely thank the reviewers who acknowledged the novelty of our class-aware prompt pool design and the parameter efficiency of our method. Your precious recognition has enhanced the visibility of the innovation in this paper.

Second, regarding the concerns raised by the reviewer **er6C** and **m3Qy** about the evaluation on real-world benchmarks (Cityscapes-to-ACDC), we conducted supplementary experiments and demonstrated that our method outperforms state-of-the-art methods like DPCore by 1.1% on this dataset. In addition, for the reviewer **8gyU** and **TUaD**'s questions regarding efficiency, we supplemented more comprehensive performance analyses focused on efficiency metrics, including runtime, GPU memory usage and computational cost (FLOPs), to systematically compare our method with baselines and clarify the performance-efficiency trade-off. With these responses, Reviewer **8gyU** acknowledged our comprehensive evaluation and accuracy improvements, stating that they would increase their score. At the same time, other reviewers had no further questions after our thorough explanations. We deeply appreciate all reviewers' efforts, which have further improved the quality of our paper.

In summary, the suggestions raised by the reviewers do not fundamentally affect our core ideas, methodological innovations, or experimental analyses. Nevertheless, we commit that all reviewers' comments are of great importance and will be fully reflected in the final version.

Finally, we would like to once again extend our most sincere thanks to all reviewers, area chairs, and program chairs!

Sincerely,

The Authors

---

### Decision · Program_Chairs · 2025-09-17

**Decision:**

Accept (poster)

**Comment:**

The paper proposes a continual test-time adaptation method that separates new domain knowledge from old knowledge to reduce interference across the domains, and later merges them to control complexity and retain useful information. Reviewers in general agreed that the paper is clearly written, the tackled problem is important, and that the proposed method shows strong performance across standard corrupted datasets as well as more realistic domain-shift scenarios, while remaining efficient. While there were some concerns raised by the reviewers, the rebuttal sufficiently addressed them with additional experiments, sensitivity analyses, and comparisons to additional baselines, clarifying robustness and efficiency trade-offs, which led to all reviewers leaning toward acceptance. Yet, there were remaining concerns regarding too many hyperparameters, limited novelty and efficiency improvement over existing works. Overall, the work makes steady progress on a timely problem, with thorough experiments and solid improvements, making it a borderline accept case.